# Social isolation modulates appetite and avoidance behavior via a common oxytocinergic circuit in larval zebrafish

Caroline L. Wee [1,2,3,5], Erin Song [1,5], Maxim Nikitchenko[1,4,5], Kristian J. Herrera [1], Sandy Wong[1], Florian Engert [1,6✉] & Samuel Kunes[1,6✉]

Animal brains have evolved to encode social stimuli and transform these representations into advantageous behavioral responses. The commonalities and differences of these representations across species are not well-understood. Here, we show that social isolation activates an oxytocinergic (OXT), nociceptive circuit in the larval zebrafish hypothalamus and that chemical cues released from conspecific animals are potent modulators of this circuit's activity. We delineate an olfactory to subpallial pathway that transmits chemical social cues to OXT circuitry, where they are transformed into diverse outputs simultaneously regulating avoidance and feeding behaviors. Our data allow us to propose a model through which social stimuli are integrated within a fundamental neural circuit to mediate diverse adaptive behaviours.

[1] Department of Molecular and Cellular Biology and Center for Brain Science, Harvard University, Cambridge, Massachusetts, USA. [2] Program in Neuroscience, Department of Neurobiology, Harvard Medical School, Boston, Massachusetts, USA. [3] Present address: Institute of Molecular and Cell Biology, A*STAR, Singapore. [4] Present address: Duke University, Durham, North Carolina, USA. [5] These authors contributed equally: Caroline L. Wee, Erin Song, Maxim Nikitchenko. [6] These authors jointly supervised this work: Florian Engert, Samuel Kunes. ✉email: florian@mcb.harvard.edu; kunes@fas.harvard.edu

In mammals, signaling in oxytocinergic (OXT) circuits modulates a wide spectrum of socially-driven behaviors, ranging from pair bonding and parental care to responses to stress and pain[1–3]. OXT has also been described as a potent appetite suppressant[4,5]. We and others previously reported that larval zebrafish OXT neurons encode a response to aversive, especially noxious stimuli, and directly drive nocifensive behavior via brainstem premotor targets[6,7]. Moreover, studies in both zebrafish[6–8] and mammals[9,10] suggest that the highly conserved[11] OXT-expressing neuronal population is anatomically and functionally diverse and might modulate multiple behaviors.

The optically transparent and genetically tractable larval zebrafish is a popular model for relating brain activity to behavioral output. Social-cue sensing (visual and mechanosensory) and social interactions have been reported at these larval stages, although they tend to manifest as avoidance behaviors, with social attraction only developing after 2 weeks of age[12–15]. One exception is in the domain of olfaction, where olfactory detection and imprinting of chemical kin cues in zebrafish and other teleost fish have been reported as early as 6 days post-fertilization (dpf)[16], with animals already demonstrating a preference for kin chemical cues at these early ages[17,18]. Recent work suggests that major histocompatibility complex (MHC) signaling via specific olfactory sensory neuron subtypes (e.g., crypt cells) could allow for the recognition of kin olfactory signals, and a downstream forebrain region, the subpallium, has been proposed to mediate such affiliative social behaviors[16–21].

Here, in a brain-wide screen[22] for neuronal populations whose activity reflects social context, we show that larval zebrafish oxytocinergic neurons display diverse responses to conspecific chemosensory stimuli and are key effectors for social-context modulation of nociceptive and appetite-driven behaviors. We further describe an olfactory bulb to subpallium circuit that is responsive to conspecific cues and sufficient to modulate OXT neuronal activity. Our results reveal a simple algorithm by which neuromodulatory neurons can represent social context to exert flexible control over hard-wired behavioral drives.

## Results

### Brain-wide activity mapping of social isolation and its rescue by conspecific chemical cues.

Using phosphorylated extracellular signal-regulated kinase (pERK)-based whole-brain activity mapping (MAP-Mapping[22]), neural activity in brains of briefly (2 hrs) socially isolated larvae (7–8 dpf) was compared with animals that had been maintained in the presence of similarly aged conspecifics. We found that isolated fish showed an enhancement of neural activity in specific regions, including the telencephalon, hindbrain, locus coeruleus, area postrema, caudal hypothalamus, preoptic area (PO, homolog of the hypothalamic paraventricular nucleus in mammals), and posterior tuberculum (PT) (Fig. 1a, b, Supplementary Movie 1, Supplementary Data 1). Many of these same regions were observed to be activated by noxious or aversive stimuli[6,7] and may thus represent the signatureactivity pattern of a negative internal state that can be likewise triggered by social deprivation. Further, these regions appear to scale their activity in a monotonic fashion with the negative valence of the stimulus[6] (Fig. 1a), and might encode the state of social isolation analogously.

Neurons expressing the peptide oxytocin (OXT) are localized to the PO and PT[6,23,24] regions. Indeed, by measuring pERK activity for individual GFP-labeled OXT neurons ($Tg(oxt:GFP)$) in high-resolution confocal microscopic images of dissected brains[6] (Fig. 1b–d, Supplementary Fig. 1), we found that OXT-positive neurons clustered in both the preoptic area ($OXT_{PO}$) and posterior tuberculum ($OXT_{PT}$) display significantly enhanced activity in socially isolated fish. To further elucidate the nature of this social signal, we compared OXT neuronal activity in animals exposed to the separated visual or chemical cues of a social environment. Water conditioned by prior exposure to conspecific larval fish (see Methods) reduced the elevated $OXT_{PO/PT}$ neuronal activity observed in socially isolated fish, whereas visual exposure to conspecific larval fish (maintained in a separate water enclosure) was less effective in reducing $OXT_{PO}$ neural activity (Fig. 1c, Supplementary Fig. 1a). We also examined the effect of water conditioned with similarly aged sibling (kin) fish in relation to similarly aged fish of a different strain background (non-kin). Both "kin" and "non-kin" conditioned water were sufficient to reduce $OXT_{PO}$ neuron activity (Fig. 1c, Supplementary Fig. 1a), but the effect of kin water was overall stronger (Fig. 1c). Thus, our results show that hypothalamic OXT activity is increased during brief (2-hr) social isolation and that this enhanced activity is significantly (and rapidly, see Fig. 2) suppressed by social cues. The $OXT_{PO}$ cluster displays a particular preference to chemical (and non-visual) cues in water conditioned by conspecific larval fish (Fig. 1c–e, Supplementary Fig. 1a). $OXT_{PT}$ neurons on the other hand showed stronger suppression by visual and non-kin cues (Fig. 1c–e, Supplementary Fig. 1a). Further, nearby OXT-negative neurons in the PO and PT area were also suppressed by conspecific-conditioned cues, including visual cues, suggesting that they too may play a role in integrating social responses (Supplementary Fig. 1b, c).

### In vivo calcium imaging reveals diverse OXT neuronal responses to chemical conspecific cues.

We next turned to in vivo calcium imaging to acquire a more temporally precise record of OXT neuronal activity. Imaging was performed on larvae in which $Tg(UAS:GCaMP6s)$ was driven directly in OXT neurons with a $Tg(oxt:Gal4)$ driver (Fig. 2a). Tethered 8–11-dpf fish were subjected to either conspecific-conditioned or control (unconditioned) water released in 10-second pulses. We found that control water flow presents a mechanosensory stimulus that mildly activates OXT neurons, whereas kin-conditioned water triggers an immediate reduction of OXT neuronal activity (Fig. 2a, b). We did not observe any difference in the effectiveness of kin-conditioned water with respect to social isolation time (from 30 min to 4 h) or age range (8–11 dpf) (Supplementary Fig. 2a, b). We next examined whether OXT neuron calcium activity displays distinctions with respect to kin and non-kin conspecific water-borne cues[16,19]. Similar to our pERK experiments in Fig. 1, OXT neuronal responses were compared between sibling-conditioned water ("kin water") and water conditioned by larvae of a distinct genetic background[25] ("non-kin water"). To minimize the effects of familiarity, conspecifics used to generate conditioned cues were raised apart from experimental fish. Kin water was found to induce a greater reduction in $OXT_{PO}$ activity compared with non-kin water. We also examined water conditioned by incubation with adult kin, which is a potential aversive cue since adult zebrafish consume their own young[26,27]. In contrast to larval-conditioned water, adult-conditioned water induced a substantial increase in $OXT_{PO/PT}$ activity.

OXT neurons are diverse in their circuit connectivities[8,28–30] and responses to social isolation (Fig. 1) and nociceptive input[6]. To further resolve this heterogeneity, we classified individual OXT neurons into populations that showed either reduced (blue) or increased (red) calcium activity in response to water-borne conspecific cues (Fig. 2c–e). Across a range of thresholds, the fraction of OXT-positive neurons whose activities were suppressed by larval kin-conditioned water was consistently higher than the fraction of neurons that were activated (Supplementary Fig. 2c). Non-kin-conditioned water, in contrast, induced equivalent neuronal fractions with increased or suppressed activities (Supplementary Fig. 2c). Water conditioned by adult fish triggered a greater fraction of neurons with enhanced activity

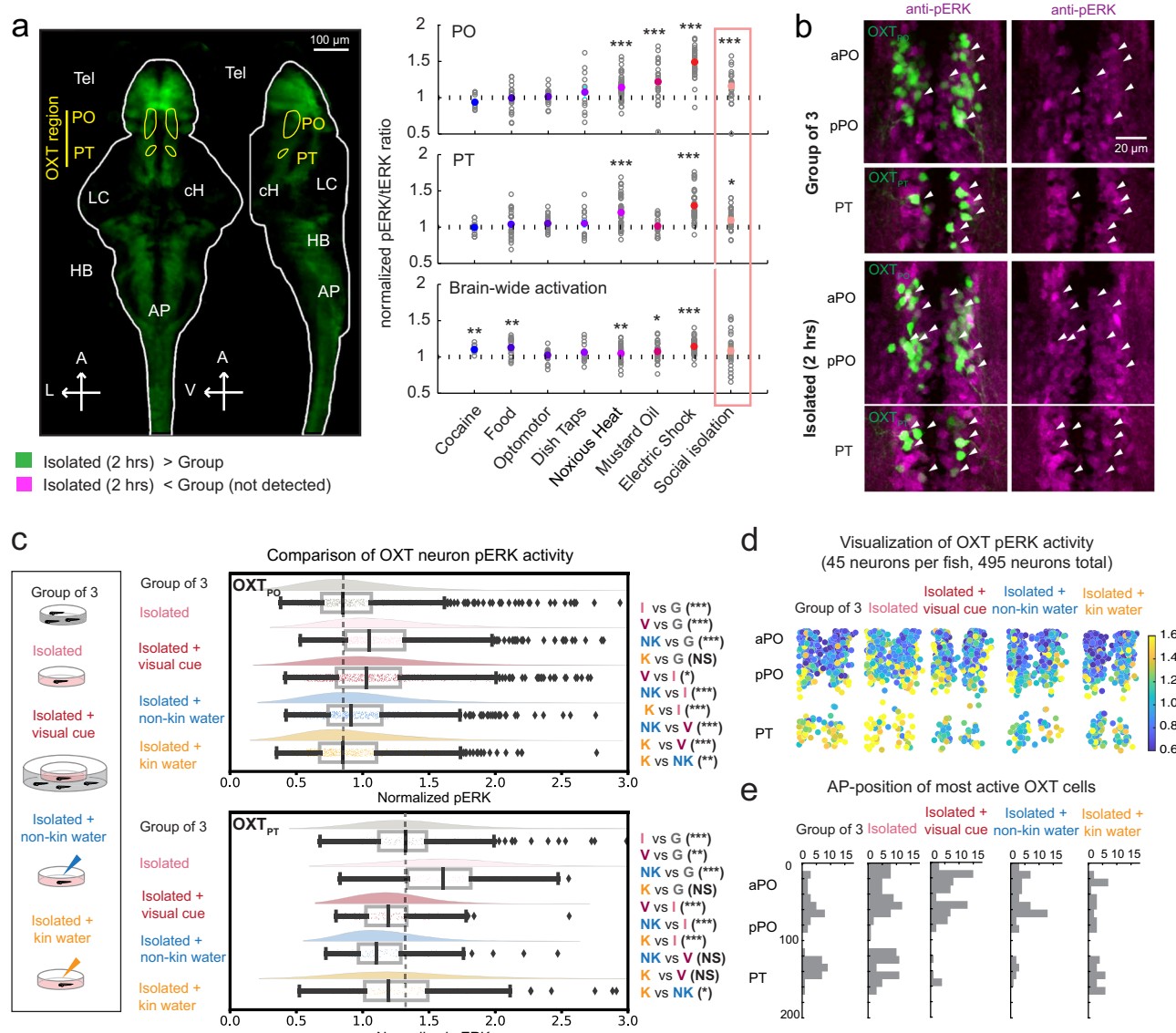

**Fig. 1 Phospho-ERK-based mapping reveals OXT neuron modulation by social context. a** Left: MAP-mapping[22] of 2-hr isolated fish vs fish in groups. Green voxels indicate significantly higher activity. Map combines data from 5 experiments in which isolated fish are compared with groups of 3 (2 experiments), 5 (1 experiment), or 10 (2 experiments). Yellow = outline of preoptic (PO) and posterior (PT) OXT populations used to quantify activity in Right. Tel = telencephalon, cH caudal hypothalamus, LC locus coeruleus, AP area postrema, HB hindbrain. A anterior, L left, V ventral. Scale bar = 100 μm. Right: Social isolation significantly activates PO and PT regions. Data for other stimuli were also included in Wee et al. (2019)[6]. Adjusted p-values for social isolation: ***$p = 0.00084$ (PO), *$p = 0.03$ (PT), $p = 0.42$ (whole brain), two-sided Wilcoxon signed-rank test relative to a median of 1, Bonferroni correction. **b** Maximum-intensity projection images showing pERK expression (magenta) in *Tg(oxt:GFP)*-positive (green) and surrounding neurons, from a representative dissected brain of an isolated fish (bottom) and a fish kept in groups of 3 (top). White arrows indicate examples of OXT neurons with high pERK intensities. Scale bar = 20 μm. This experiment was repeated more than 5 times with similar results. **c** Fish were either kept in groups of 3 (gray, $n = 835$ OXT$_{PO}$/158 OXT$_{PT}$ neurons from 12 fish), isolated (pink, $n = 803$ OXT$_{PO}$/128 OXT$_{PT}$ neurons from 11 fish), isolated but exposed to visual cues of conspecifics (red, $n = 796$ OXT$_{PO}$/106 OXT$_{PT}$ neurons from 12 fish), or isolated but exposed to non-kin-conditioned water (blue, $n = 751$ OXT$_{PO}$/115 OXT$_{PT}$ neurons from 11 fish) or kin-conditioned water (orange, $n = 722$ OXT$_{PO}$/134 OXT$_{PT}$ neurons from 11 fish). Boxplot shows the median (center), interquartile range (IQR; box), 1.5 IQRs of the lower and upper quartile (whiskers), and outliers beyond this range (diamonds); Half-violin plot shows kernel-density estimate of normalized pERK values. OXT$_{PO}$ neurons: adjusted $p = 0$*** (group vs isolated)/0*** (group vs visual)/$1.7 \times 10^{-4}$*** (group vs non-kin water)/1 (group vs kin water)/0.023* (isolated vs visual)/$3.4 \times 10^{-18}$*** (isolated vs non-kin water)/0*** (isolated vs kin water)/$4.8 \times 10^{-8}$*** (visual vs non-kin water)/$4.2 \times 10^{-22}$***(visual vs kin water)/ 0.0014** (kin vs non-kin water). Kruskal–Wallis Test with Tukey–Kramer correction for multiple comparisons. OXT$_{PT}$ neurons: adjusted $p = 7.9 \times 10^{-6}$*** (group vs isolated)/0.0071** (group vs visual)/$4.3 \times 10^{-6}$*** (group vs non-kin water)/0.20 (group vs kin water)/$1.1 \times 10^{-13}$*** (isolated vs visual)/$6.8 \times 10^{-21}$*** (isolated vs non-kin water)/$1.0 \times 10^{-10}$*** (isolated vs kin water)/0.58 (visual vs non-kin water)/0.69 (visual vs kin water)/0.032* (kin vs non-kin water). Kruskal–Wallis test with Tukey–Kramer correction for multiple comparisons. **d** Spatial distribution of OXT neurons from each category, color coded according to normalized pERK intensity (most active = yellow, least active = deep blue, colorbar shows normalized pERK value). See Methods. **e** Anterior–posterior (AP) localization of "active cells" sampled from each category. See Methods. Source data are provided as a Source Data file.

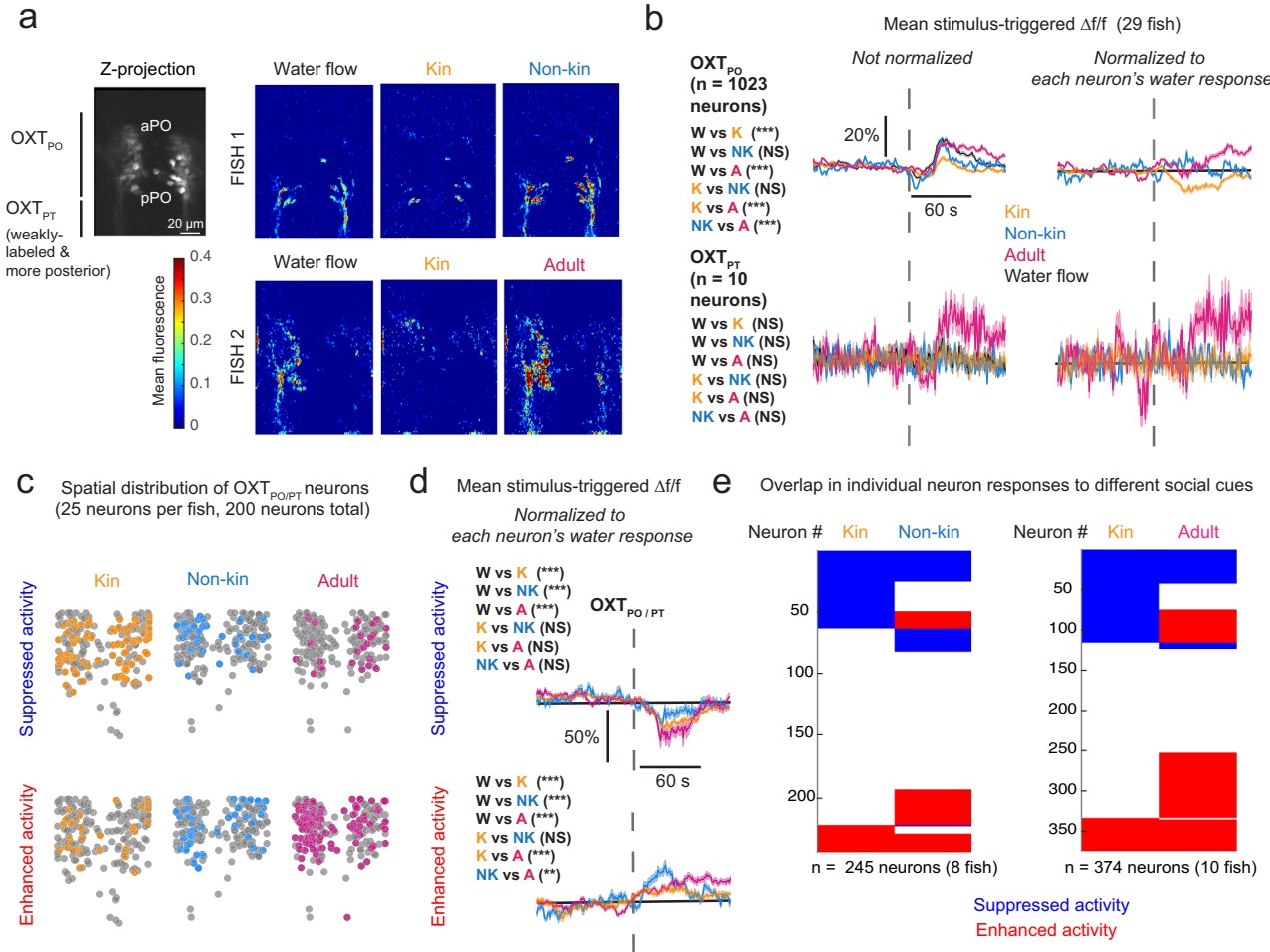

**Fig. 2 Calcium imaging confirms OXT neuron modulation by conspecific cues. a** Left: Maximum z-projection of oxytocin neurons. The *Tg(oxt:Gal4; UAS:GCaMP6s)* transgenic line labels OXT$_{PO}$ neurons strongly, with weaker labeling of the more ventro-posteriorly located OXT$_{PT}$ cluster. Scale bar = 20 μm. Right: Mean OXT neuron fluorescence over a 60-s period post stimulus from two different fish, one imaged with kin and non-kin water (top), and another with kin and adult water (bottom). This experiment was repeated on 8 and 10 fish, respectively, with similar results. **b** Left: Mean stimulus-triggered calcium activity (Δf/f) of OXT$_{PO}$ (top) and OXT$_{PT}$ (bottom) neurons. All fish (29 fish, n = 1033 neurons) were imaged with water or kin water, some also with either non-kin (8 fish, n = 245 neurons) or adult water (10 fish, n = 374 neurons). Of all neurons imaged, only 10 were from the OXT$_{PT}$ population. Right: Mean stimulus-triggered Δf/f of each neuron normalized to its own mean water-flow response. Gray broken line indicates stimulus onset. Shading indicates SEM. Asterisks show statistical comparison of mean Δf/f over a 60-s post-stimulus period for different cues, two-sided Kruskal–Wallis Test with Tukey–Kramer correction. OXT$_{PO}$: ***p = 1.4 × 10$^{-9}$ (water vs kin)/p = 0.13 (water vs non-kin)/***p = 4.0×10$^{-11}$ (water vs adult)/p = 0.30 (kin vs non-kin)/***p = 0 (kin vs adult)/***p = 2.5 × 10$^{-11}$ (non-kin vs adult). OXT$_{PT}$: p = 0.99 (water vs kin)/p = 0.76 (water vs non-kin)/p = 0.97 (water vs adult)/p = 0.85 (kin vs non-kin)/p = 1 (kin vs adult)/p = 0.95 (non-kin vs adult). **c** Spatial distribution and percentages of neurons that show either suppressed (top) or enhanced (bottom) responses to each cue relative to water. See Methods and Supplementary Fig. 2c for details. **d** Mean stimulus-triggered Δf/f of neurons classified as being suppressed (top) or enhanced (bottom) by each cue, normalized to their mean water-flow response. Shading indicates SEM. Asterisks show statistical comparison of mean Δf/f over a 60-s post-stimulus period for different cues, two-sided Kruskal–Wallis test with Tukey–Kramer correction. For water responses, only neurons that had suppressed or enhanced activity induced by any of the other cues were included in the statistical analysis. Suppressed activity: ***p = 0 (water vs kin)/ ***p = 1.3 × 10$^{-7}$ (water vs non-kin)/***p = 8.2 × 10$^{-7}$ (water vs adult)/p = 1 (kin vs non-kin)/p = 0.71 (kin vs adult)/p = 0.84 (non-kin vs adult); n = 429 (water)/390 (kin)/47 (non-kin)/65 (adult). Enhanced activity: ***p = 6.7 × 10$^{-14}$ (water vs kin)/***p = 2.8 × 10$^{-10}$ (water vs non-kin)/***p = 0 (water vs adult)/p = 0.88 (kin vs non-kin)/***p = 8.5 × 10$^{-7}$ (kin vs adult)/**p = 0.0043 (non-kin vs adult); n = 265 (water)/117 (kin)/55 (non-kin)/116 (adult). **e** Left: overlap between kin and non-kin water responses. Right: overlap between kin and adult water responses. Red = enhancement, Blue = suppression, White = no change. Kin cues induced the highest percentage of suppression (25–30%) and lowest percentage of activation (9–11%), whereas adult cues induced the lowest percentage of OXT neuron suppression (14%) and highest percentage of activation (46%) of OXT neurons. Non-kin cues induced an intermediate level of OXT neuron suppression (18%) and activation (23%). See Supplementary Fig. 2d, e for additional visualization. Source data are provided as a Source Data file.

(Supplementary Fig. 2c). Under all conditions, activated or suppressed neurons were spatially distributed instead of being segregated into distinct areas (Fig. 2c). Further, the specific response properties of a neuron to one conspecific cue did not strongly predict how it would respond to the other cues, although we observed that kin water-activated OXT neurons tended to also be commonly activated by other conspecific (non-kin or adult)

cues (Fig. 2e, Supplementary Fig. 2d, e). The response heterogeneity of the OXT population suggests that these neurons differentially encode conspecific stimuli.

**The zebrafish olfactory bulb and downstream subpallium encode conspecific chemical cues.** Since water-borne chemical

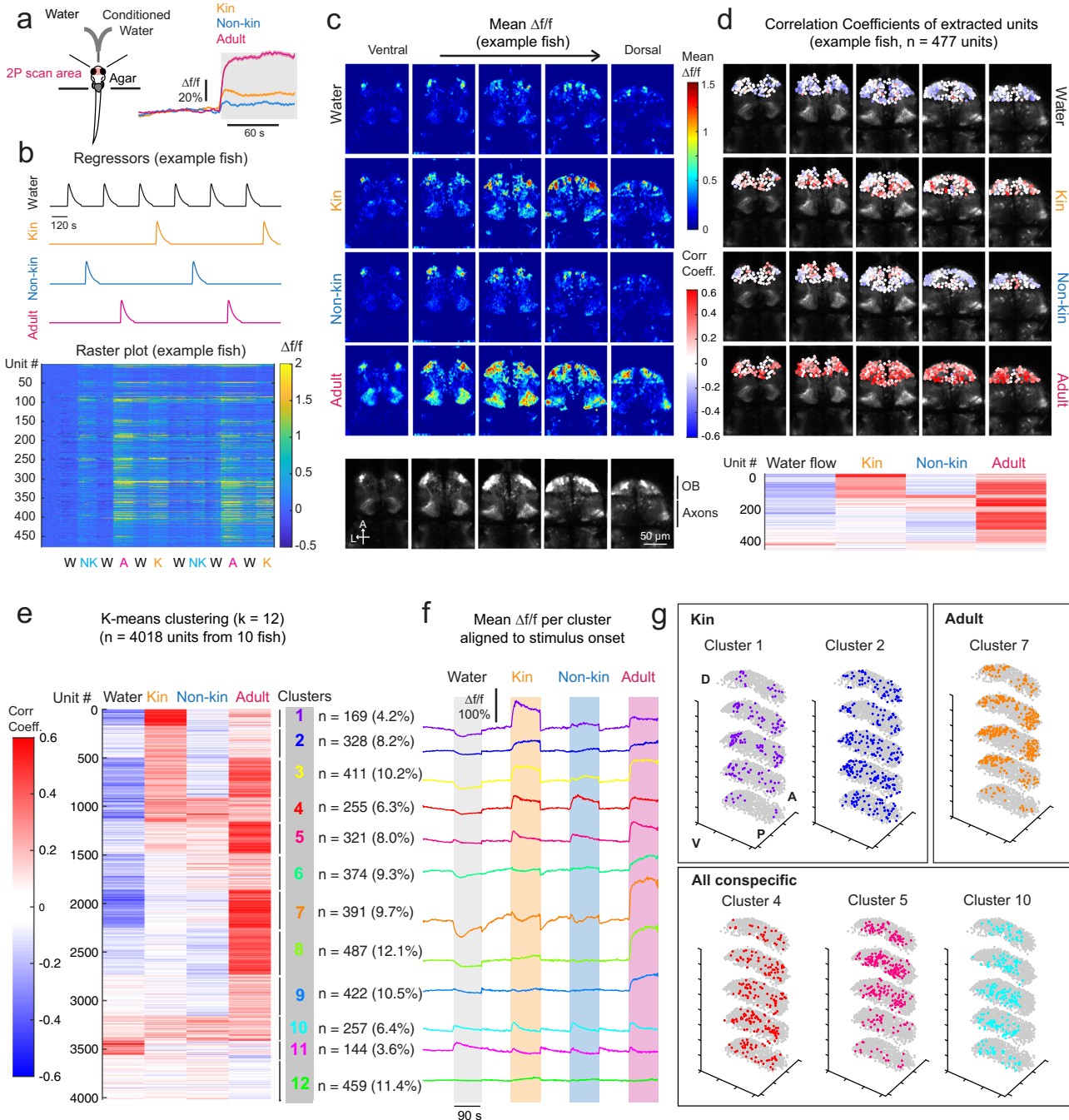

cues might act as odorants, we included the olfactory bulb (OB) in the areas specifically interrogated for changes in neuronal activity upon exposure to chemical conspecific cues that modulate OXT neuron activity. We found that each of three conditioned water types (kin, non-kin, and adult kin) generated overlapping but distinct activity signatures within the OB (Fig. 3a–c, Supplementary Fig. 3). Of all the cues, adult water had the strongest and longest-lasting effect (Fig. 3a, b), as was also noted for OXT neuronal responses with this stimulus (Fig. 2b). Notably, non-kin water at the same concentration elicited much weaker OB activity than kin water, consistent with the prior observations of its weaker effect in suppressing OXT neuron activity (Figs. 1c, d, 2, 3c).

To characterize the relationships in OB encoding of the different social stimuli, we extracted individual units[31] and performed K-means clustering (k = 12) based on their correlation coefficients with stimulus regressors (Fig. 3d–f, see Methods). We

identified clusters that were exclusive to specific social cues as well as clusters with mixed selectivity. Consistent with previous reports[20], kin-selective (as well as adult-selective clusters) tended to be located more laterally, while nonselective social-cue-responsive clusters were situated medially (examples shown in Fig. 3g; all clusters shown in Supplementary Fig. 3c).

The zebrafish telencephalon contains multiple GABAergic populations that receive olfactory input[21,32,33] and hence might transform excitatory input from the OB into inhibitory signals to the OXT population. We focused on the subpallium (SPa) in particular because it is a known downstream target of the olfactory bulb[21,32,33] and implicated in social behavior[34,35] and kin odor recognition[21]. To this end, we imaged neuronal activity in the telencephalon in the pan-neuronal Tg(HuC:GCaMP6s) line in which GABAergic neurons were also fluorescently marked with red fluorescent protein (Tg(Gad1b:RFP); Supplementary

**Fig. 3 Calcium imaging reveals olfactory bulb encoding of conspecific stimuli. a** Left: Schematic of imaging and cue-delivery setup. Two-photon (2-P) calcium imaging of the olfactory bulb was performed as each *Tg(HuC:GCaMP6s)* fish was presented with pulses of kin water, non-kin water, and adult water (randomized order), alternated with pulses of water. Right: Stimulus-triggered average of all olfactory bulb (OB) neuron responses (Δf/f) to kin, non-kin, and adult water. **b** Δf/f traces (bottom, raster plot for an example fish) from individual units (see Methods) within the OB were correlated with regressors (top) fit to each stimulus type. **c** Mean OB Δf/f integrated over a 60-s period post stimulus from a single fish (same fish shown in Fig. 3b, also see Supplementary Fig. 3), shown for each plane (ventral to dorsal) and each cue in a volumetric stack. Bottom-most panel shows the anatomy references for each plane. A anterior, L left. Scale bar = 50 μm. This experiment was repeated on 10 fish with similar results. **d** Δf/f traces from individual units were extracted from the OB (see Methods). Top: Correlation coefficients for each unit (*n* = 477 units from the same fish in **b** and **c**) with each regressor are color coded (blue to red) and overlaid over the anatomy image per z-plane. Bottom: Each unit's response (i.e., correlation coefficient) to water and conspecific stimuli, from the same fish as above, clustered using K-means clustering (as in **e**). **e** K-means clustering (k = 12) was performed on the matrix of correlation coefficients (color coded blue to red) to water and conspecific stimuli (*n* = 4018 units from 10 fish). Number of units within each cluster and percentage representation of total units are displayed on the right. Clusters specific to kin water and adult water, as well as clusters with mixed selectivity were observed. **f** Mean stimulus-triggered Δf/f for each cluster, aligned to stimulus onset. Order of clusters is the same as in (**e**). As OB responses are often sustained across the entire stimulus epoch, negative Δf/f and correlation coefficients in response to water flow are likely due to washout of residual olfactants, rather than an active suppression of the signals by water flow. **g** 3D plot displaying the spatial localization of units within each cluster, for a few selected clusters (all clusters shown in Supplementary Fig. 3c). XY coordinates for all units per fish were scaled linearly to their minimum and maximum values in each dimension. Adult-responsive clusters tend to be situated laterally (e.g., cluster 7), whereas clusters that show more general conspecific responses (e.g., clusters 4,5,10) are situated more medially. Kin-specific responses (clusters 1 and 2) are more scattered, though they extend more laterally than general conspecific-responsive clusters. A anterior, P posterior, D dorsal, V ventral. Source data are provided as a Source Data file.

Figs. 4, 5). We also imaged activity in the preoptic area of fish in which OXT neurons are labeled with mCherry (Supplementary Fig. 5). The SPa can be broadly divided into anterior (aSPa) and posterior (pSPa) domains; both of these subpallial domains, including their GABA-positive units, were responsive to conspecific cues (Supplementary Figs. 4, 5). Using K-means clustering as above (k = 12), we identified neuronal clusters responsive to distinct social cues, notably with more kin- and adult kin-responsive clusters compared with non-kin-responsive clusters (Supplementary Figs. 4, 5). In contrast to the SPa, larval conspecific cues reduced $OXT_{PO}$ neuronal activity (Supplementary Fig. 5). These observations are consistent with a role of the SPa as an intermediary in the olfactory modulation of OXT neuronal activity.

**A GABAergic subpallial population encodes conspecific chemical cues and suppresses oxytocin neuron activity.** To demonstrate a circuit pathway through which OB-encoded social cues might be conveyed to the OXT population, we sought Gal4 transgenic lines with which to perturb neuronal activity in the subpallium (Fig. 4a). The *Tg(y321:Gal4)* enhancer trap specifically labels the aSPa population and has been previously implicated in visually mediated social behavior and social behavior development in larval zebrafish[34–36]. Co-labeling y321:Gal4-positive and Gad1b-positive cells in double transgenics revealed that, among y321-labeled neurons, 88% in the aSPa ($aSPa^{y321}$) and 87% in the pSPa ($pSPa^{y321}$) are GABAergic (*n* = 7 fish, Fig. 4b). Moreover, some OXT neurons extend dendrites into the SPa, establishing potential connectivity with the $SPa^{y321}$ population (Fig. 4c). Analysis of activity via pERK staining (similar to Fig. 1) revealed stronger responses of both $aSPa^{y321}$ and $pSPa^{y321}$ neurons to kin water, as compared with non-kin water (Supplementary Fig. 6a, b). In addition, calcium imaging of the $aSPa^{y321}$ neurons' response to conspecific cues (Fig. 4d–f) revealed clusters of $aSPa^{y321}$ neurons that are differentially responsive to social cues, with more clusters responsive to kin and adult water-borne cues, consistent with results from pan-neuronal and GABAergic neuronal activity imaging (Supplementary Figs. 4, 5).

These correlative results suggested a possible mechanism by which kin cues are transformed by subpallial GABA neurons into an inhibitory signal that suppresses OXT neuronal activity. To establish a functional relationship between these two populations, we optogenetically stimulated y321:Gal4-positive neurons via

spatially restricted laser-pulse illumination of red-shifted channelrhodopsin (ReaChR[37]) and concurrently recorded calcium sensor (GCaMP6s) fluorescence in OXT neurons (Fig. 5a). In control experiments, with sibling fish lacking the ReaChR transgene, laser illumination of the subpallium induced a small increase in y321:Gal4 neuronal activity (Fig. 5b, c, top rows, Supplementary Fig. 6c), indicating that there is some calcium response in the absence of ReaChR. When both the ReaChR and y321:Gal4 transgenes were present, y321-positive neuronal activity was strongly induced and phase-locked to the onset of the laser pulse, confirming the efficacy of the optogenetic protocol (Fig. 5b).

We then examined OXT neuronal activity after spatially targeted aSPa illumination in fish harboring the y321:Gal4, oxt:Gal4, UAS:ReaChR-RFP, and UAS:GCaMP6s transgenes. These animals express ReaChR and GCaMP6 in both the preoptic OXT population and the y321-positive subpallium neurons. OXT neuronal activity in these animals was compared with control siblings lacking the y321:Gal4 transgene, hence lacking ReaChR expression in the subpallium (Fig. 5b, c). In these negative-control fish, laser stimulation of the subpallium was associated with sporadic OXT neuronal activity and moderate reductions in OXT neuropil activity (blue arrows in bottom traces, Fig. 5b). We consider it possible that these effects reflect optogenetic activation of ReaChR in OXT neuron dendrites that extend into the subpallium (Fig. 4c). In contrast, with y321:Gal4 present and ReaChR expressed in subpallial neurons, optogenetic stimulation of the aSPa resulted in a dramatic and significant reduction in OXT neuronal activity, specifically in $OXT_{PO}$ cell bodies and the OXT neuropil (OXT axons) (Fig. 5b, c, Supplementary Fig. 6d). Thus, our data strongly suggest that GABAergic $aSPa^{y321}$ neurons transform excitatory input from conspecific cue activation of the olfactory bulb into an inhibitory signal for OXT neuronal activity.

**Conspecific cues suppress nociceptive OXT circuits and defensive behavior.** Social buffering is a widely observed phenomenon in which the presence of conspecifics ameliorates the effects of aversive experience[38,39]. We previously showed that a large fraction of OXT neurons are activated by noxious stimuli and drive defensive behaviors, specifically through brainstem premotor targets that trigger vigorous large-angle tail bends[6]. Given that most of the OXT circuitry appears to be suppressed by

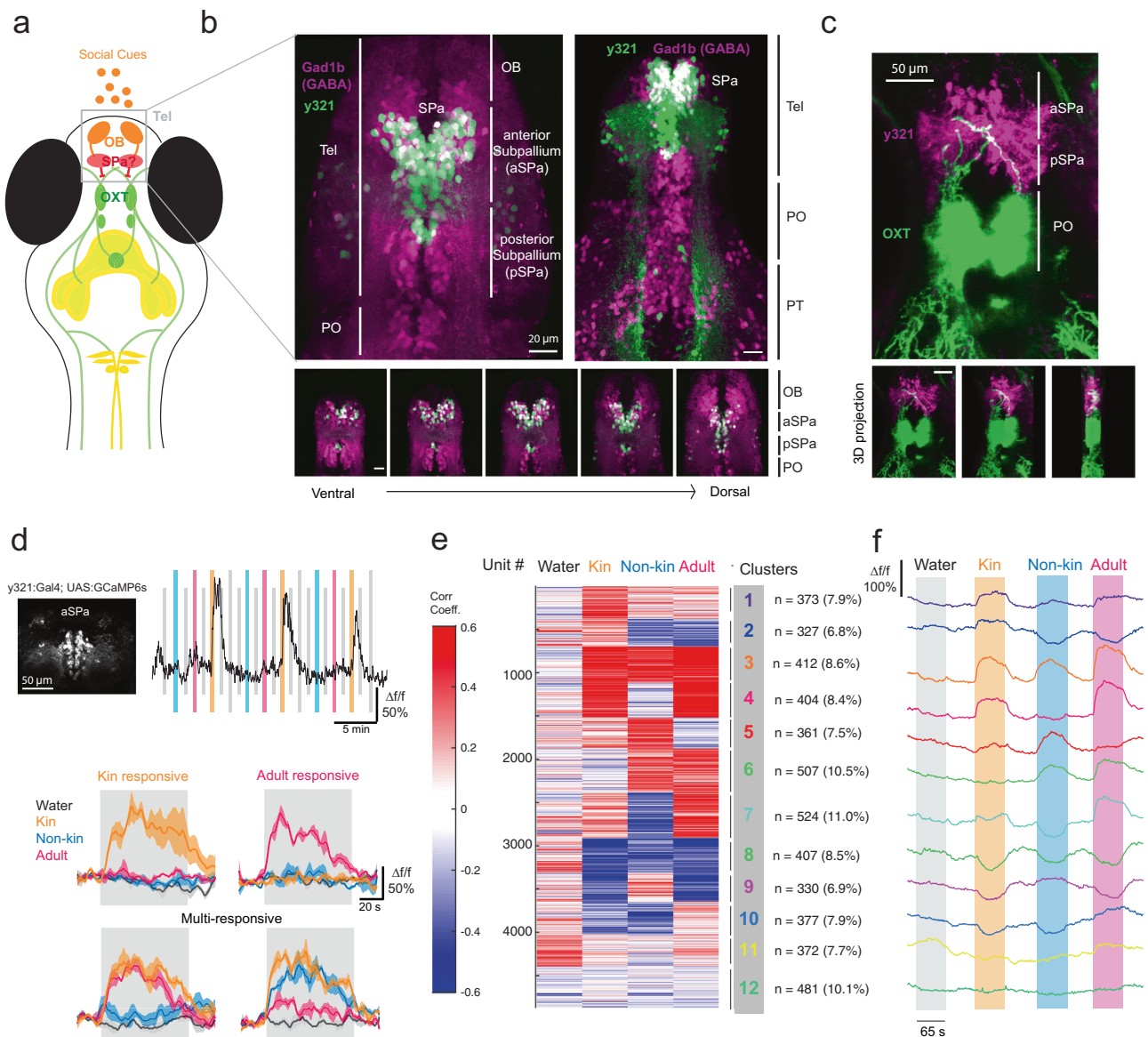

**Fig. 4 A genetically defined GABAergic subpallial population discriminates social cues. a** Schematic depicting the hypothesized telencephalic subpallial (SPa) pathway linking olfactory bulb conspecific cue responses to preoptic OXT neurons. **b** Top left: Maximum-projection image showing overlap of *Tg(y321:Gal4;UAS:Kaede)* (green) with GABAergic neurons labeled by *Tg(Gad1b:RFP)* (magenta) in the telencephalon. y321 and Gad1b co-expression occurs in both the anterior subpallium (aSPa, rostral to the anterior commissure) and posterior commissure (pSPa, caudal to the anterior commissure and rostral to the preoptic area). Top right: Lower-magnification maximum-projection image showing overlap of *Tg(y321:Gal4;UAS:Kaede)* with *Tg(Gad1b:RFP)* (magenta). Overall, 88.2 ± 1.5% of aSPa$^{y321}$ and 87.3 ± 3.06% of pSPa$^{y321}$ neurons are GABAergic ($n = 7$ fish). OB olfactory bulb, Tel telencephalon, PO preoptic area; PT posterior tuberculum. Bottom: y321 and Gad1b overlap shown across different z-planes. All scale bars = 20 μm. This experiment was repeated on 7 fish with similar results. **c** OXT neuron dendrites project into the location where SPa$^{y321}$ neurons reside. Top: Maximum-projection image; Bottom: 3D projection at different rotation angles. Scale bar = 50 μm. This experiment was repeated on more than 3 fish with similar results. **d** 2-P calcium imaging was performed on *Tg(y321:Gal4; UASGCaMP6s)* neurons in the aSPa region on fish exposed to kin, non-kin, and adult water cues. See Supplementary Figs. 4, 5 for imaging and analysis of the same telencephalic regions in *Tg(HuC:GCaMP6s; Gad1b:RFP fish)*. Top left: Maximum-projection image showing the aSPa$^{y321}$ region imaged. Scale bar = 50 μm. Top right: Example Δf/f trace of a kin-responsive unit. Bottom: Stimulus-triggered averages for units that are kin-selective, adult-selective, or responsive to multiple conspecific cues. Shaded region indicates SEM. This experiment was repeated on 4 fish with similar results. **e** K-means clustering ($k = 12$) was performed on the matrix of correlation coefficients (color coded blue to red) to water and conspecific stimuli $n = 4875$ units from 4 fish. Number of units within each cluster and percentage representation of total units are displayed on the right. Clusters specific to kin, non-kin, and adult water, as well as clusters with mixed selectivity were observed. **f** Mean stimulus-triggered Δf/f for each cluster, aligned to stimulus onset. Order of clusters is the same as in (**e**). Source data are provided as a Source Data file.

water-borne social cues derived from closely related conspecifics, we posited that these cues might reduce the nocifensive behavior induced by the stimulation of TRPA1 receptors. Indeed, increased swim speed triggered by nociceptive TRPA1 receptor activation was significantly ameliorated by the presence of kin-conditioned

water (Supplementary Fig. 7). Similarly, in 8–10-dpf tethered fish, kin water significantly reduced the frequency of large-angle swims in tethered fish after kin cue delivery and TRPA1 stimulation but not during other epochs (Fig. 6a, Supplementary Fig. 8a). Correspondingly, the average calcium responses of individual OXT$_{PO}$

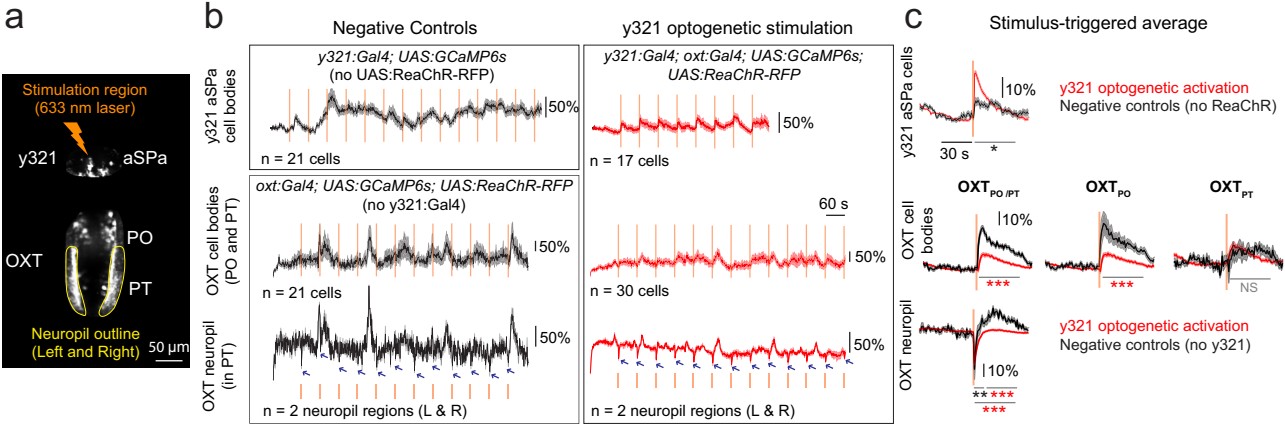

**Fig. 5 Optogenetic stimulation of subpallial neurons suppresses OXT activity. a** Maximum-projection image showing regions of optogenetic stimulation (aSPa$^{y321}$ neurons) and calcium imaging (OXT neurons in PO or PT, or neuropil bundles in PT outlined in yellow). This experiment was repeated on 18 fish with similar results. **b** Mean Δf/f traces from either aSPa$^{y321}$ neurons or OXT cell bodies/neuropil bundles after stimulation with a 633 nm laser focused on the aSPa region at 60-s intervals. Black traces depict mean Δf/f of an example ReaChR-negative-control fish (top) or y321-negative-control fish (bottom). Red traces depict mean Δf/f from an example *Tg(y321:Gal4, oxt:Gal4; UAS:GCaMP6s; UAS:ReaChR-RFP)* fish of its aSPa$^{y321}$ neurons (top), OXT$_{PO/PT}$ cell bodies (middle), or neuropil (bottom), subject to aSPa$^{y321}$ optogenetic stimulation. Orange lines indicate stimulus onset. Each box represents calcium traces from an individual representative fish. Shaded region indicates SEM. Blue arrows indicate dips in OXT neuropil activity in the presence or absence of optogenetic y321 activation. **c** Stimulus-triggered averages showing the mean Δf/f before and after the light stimulus. Responses to each stimulus were first averaged per unit (cell body or neuropil bundle) before being averaged across all units to obtain the Δf/f trace. Shaded region indicates SEM. Two-sided Wilcoxon rank-sum test was used for all statistical comparisons. Top panel: The calcium response (Δf/f) of aSPa$^{y321}$ neurons to 633 nm laser stimulation is significantly higher in the presence (red, $n = 320$ neurons from 15 fish) than absence (black, $n = 118$ neurons from 4 fish) of ReaChR (*$p = 0.031$). Middle panel: The calcium response of OXT$_{PO/PT}$ neurons under aSPa$^{y321}$ optogenetic stimulation is significantly lower (***$p = 5.4 \times 10^{-6}$, $n = 410$ neurons from 18 fish) than for negative controls (*Tg(oxt:Gal4; UAS:GCaMP6s; UAS:ReaChR-RFP)* fish) exposed to the same light stimulus ($n = 103$ neurons from 5 fish). When we separated these OXT neurons according to their anterior–posterior position, we found that OXT$_{PO}$ neurons showed significant suppression after aSPa$^{y321}$ optogenetic stimulation compared with negative controls (***$p = 0.00028$, $n = 294/91$ neurons), but no significant difference was observed for OXT$_{PT}$ neurons (OXT$_{PT}$; $p = 0.198$, $n = 116/12$ neurons). Bottom panel: OXT neuropil calcium responses under y321 optogenetic stimulation are overall significantly lower than for negative controls exposed to the same light stimulus (***$p = 2.7 \times 10^{-5}$, $n = 36/10$ neuropil regions). The dip in neuropil activity under y321 stimulation is significantly less negative than the negative controls in the first 10-s epoch (**$p = 0.0094$), whereas neuropil activity is significantly lower than the negative controls in the next 30-s epoch (***$p = 2.4 \times 10^{-5}$). Gray bars indicate the 40-s period across which Δf/f was integrated for statistical analysis; asterisk color reflects the group with the more negative (i.e., lower) integrated value. Source data are provided as a Source Data file.

(but not OXT$_{PT}$) neurons following TRPA1 stimulation were significantly reduced when preceded by kin-conditioned water (Fig. 6b).

We next examined individual OXT neuron responses to TRPA1 stimulation in order to resolve the social-cue modulation of their responses. As previously done[6], TRPA1-responsive OXT neurons were identified by correlating each neuron's calcium activity with TRPA1 stimulus and motor output regressors (Fig. 6c, Supplementary Fig. 8b). Motor regressors either reflected movements that occurred in temporal association with (within 5 s of) the TRPA1 stimulus (motor$_{stim}$) or at times outside of the stimulus period (motor$_{spon}$). For spontaneous swim bouts outside of the stimulus period, we further generated regressors for small- (motor$_{spon-S}$) and large-angle (motor$_{spon-L}$) tail bends to examine whether neurons tuned to large-angle vs small-angle movements may be differentially represented by the OXT population.

As expected, OXT neurons displayed diverse activity patterns and varying degrees of correlation with stimulus and motor regressors. Neurons tuned to the TRPA1 stimulus tended to be correlated with motor behavior that followed the TRPA1 stimulus (Fig. 6c, d, Supplementary Fig. 8c). Given the diverse responses of OXT neurons, we utilized K-means clustering (k = 12) to classify neurons on the basis of correlated activity with different regressors. This analysis uncovered 5 broad activity classes of OXT neurons (Fig. 6d): (i) TRPA1-selective neurons, with varying strength of correlation with the stimulus regressors; (ii) neurons that were both TRPA1-responsive and correlated with

spontaneous movements; (iii) neurons that were primarily motor-correlated, particularly to large-angle movements; (iv) unresponsive neurons; and (v) neurons that showed an anticorrelated relationship to the TRPA1 stimulus.

Individual OXT neurons were also grouped according to whether kin water presentation enhanced or suppressed their responses to TRPA1 stimulation (as performed in Fig. 2, see also Supplementary Fig. 8d). We found that the majority (54%) of OXT neurons displayed reduced TRPA1-induced activity in the presence of kin water, with only 32% showing enhanced activity, indicating that kin water exposure triggers widespread negative modulation of OXT neuronal responses to TRPA1 stimulation. Notably, 43% of OXT neurons already showed a reduction in activity post kin-water delivery and prior to TRPA1 stimulation, while 30% displayed increased activity. While activity suppression by kin water occurred throughout the OXT population (Fig. 6e), there were also varying ratios of suppressed versus activated neurons within each cluster and class, indicating that subsets of OXT neurons with distinct response characteristics are differentially modulated by kin water (Fig. 6e).

From the above analyses, we derive two main observations. First, moderately TRPA1-responsive OXT neurons (class i$_b$, Fig. 6e, f; 25% of OXT neurons) displayed greater activity suppression by kin water than highly TRPA1-responsive neurons (class i$_a$, Fig. 6e, f; ~3% of OXT neurons; Fig. 6e, f). However, this relationship did not extend to weakly TRPA1-responsive OXT neurons, which were not modulated by kin water (class i$_c$, Fig. 6e,

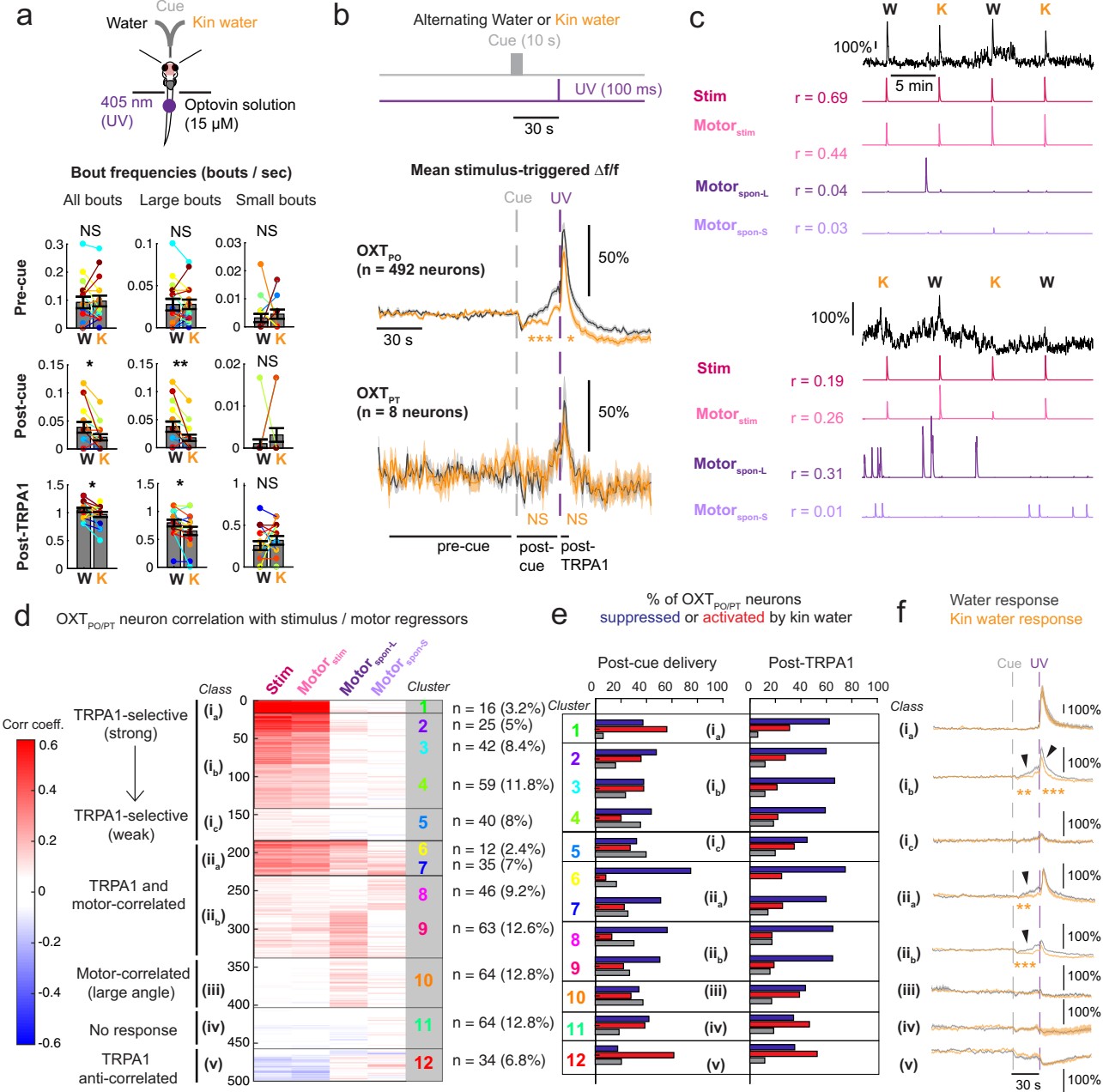

f; ~8% of OXT neurons). The average calcium responses of the moderately TRPA1-responsive (class $i_b$) neurons displayed a significant divergence from baseline activity before TRPA1 activation (Fig. 6f), suggesting that these OXT neurons respond to both water flow and TRPA1 stimulation, both of which could be significantly suppressed by kin cues. A second observation is that OXT neurons whose activity was strongly correlated with both the TRPA1 stimulus and small or large spontaneous tail movements (class ii, representing ~31% of OXT neurons) were more strongly suppressed by kin water prior to the onset of the TRPA1 stimulus. The fact that spontaneous tail movements prior to the TRPA1 stimulus were reduced (Fig. 6a) suggests these motor-correlated suppressed neurons could play the role of reducing motor responses generally in the context of an aversive cue. Overall, our results demonstrate the widespread suppressive effects of kin water on OXT neuron activity and OXT-triggered behavior. Both kin water-suppressed and activated neurons were scattered along the AP axis (Supplementary Fig. 8d), and a high

proportion were located posteriorly, indicating that at least some of the suppressed neurons have a parvocellular, hindbrain identity and may directly drive motor output[6]. Thus, the reduced nocifensive responses of larval fish observed in the presence of social cues is directly reflected in the suppressed activity of a large subset of TRPA1-responsive neurons within the OXT circuit.

**Larval conspecific water-borne cues enhance appetite via suppression of OXT neuronal activity.** Social isolation is not only associated with increases in aversive behavioral responses, but is also reportedly associated with reduced positive-valence behaviors such as feeding[40]. To further characterize the diverse behaviors subject to social modulation in larval zebrafish, we used a quantitative food-intake assay to determine whether social isolation diminishes feeding behavior[41–43] (Fig. 7a). In this assay, larval fish are deprived of food for a period of two hours, which results in increased appetite[41–43]. This appetite-modulation

**Fig. 6 Chemical kin cues modulate OXT neuron nociceptive responses and behavior. a** Top: Schematic showing setup used to probe the effect of conspecific-conditioned water on TRPA1-induced nocifensive behavior. Bottom: Kin water significantly reduced the frequency of large-angle (>50°) but not small- (≤50°) angle tail bends, both after cue delivery and post TRPA1 stimulation, though the overall bout frequency was much higher after TRPA1 stimulation (bottom-most row). Post-cue delivery: $p = 0.011^*$ (all bouts)/$0.0039^{**}$ (large angle)/0.93 (small angle). Post TRPA1: $p = 0.045^*$ (all bouts)/$0.019^*$ (large angle)/0.86 (small angle). There was no significant change in bout kinematics during epochs outside of kin cue or TRPA1 delivery ($p = 0.65/0.55/0.71$ for total, large, and small tail bends, respectively). One-sided Wilcoxon signed-rank test, $n = 16$ fish. More kinematic features are shown in Supplementary Fig. 8a. Data are presented as mean values ±SEM. **b** Top: Alternating pulses of water or conspecific water were presented, followed by a 100-ms pulse of UV light after 30 s to activate TRPA1 receptors. Half of the fish had kin water as the first stimulus. Bottom: Stimulus-triggered averages for $OXT_{PO}$ ($n = 492$) or $OXT_{PT}$ ($n = 8$) neuron calcium activity ($\Delta f/f$) in response to a water pulse or kin water delivery. The integrated $\Delta f/f$ of $OXT_{PO}$ neurons both post cue delivery ($^{***}p = 2.6 \times 10^{-12}$) and post TRPA1 stimulus ($^*p = 0.026$) was significantly lower after kin water as compared with water flow. There was no significant effect of kin water on $OXT_{PT}$ neuron activity both post cue delivery ($p = 1$) and post TRPA1 stimulus ($p = 1$), two-sided Wilcoxon signed-rank test. Black horizontal line shows the region over which the pre- and post TRPA1 calcium activity and behavior were averaged to calculate precue, post-cue, and post TRPA1 responses. Gray dashed line = water or kin water delivery, purple dashed line = UV stimulus onset, Shading indicates SEM. **c** Calcium traces ($\Delta f/f$) and the respective stimulus and motor regressors (see Results and Methods), as well as their correlation coefficients with each regressor are shown for two example $OXT_{PO}$ neurons. **d** K-means clustering ($k = 12$) was performed on the matrix of correlation coefficients to each of the stimulus or motor regressors ($n = 500$ units from 16 fish). Number of units within each cluster and percentage representation of total units are displayed on the right. Clusters specific to TRPA1 stimulation, as well as motor-correlated clusters were observed. We grouped these clusters into broader classes and subclasses (i–v), which are indicated by the black lines and boxes. See also Supplementary Fig. 8c, d. **e** The percentage of $OXT_{PO/PT}$ neurons within each cluster that were suppressed, activated, or did not show any change when exposed to kin water, either following cue delivery (left) or TRPA1 stimulation (right). Red = enhancement, Blue = suppression, White = no change. The broader classes (i–v) are indicated by black boxes. See also Supplementary Fig. 8d. **f** Mean stimulus-triggered calcium responses ($\Delta f/f$) of all $OXT_{PO/PT}$ neurons in the presence of water (black) or kin water (orange), as a function of their class/subclass. Note scale bar for class $i_a$ is different from the others due to the intensity of TRPA1 responses. For class $i_b$, the integrated calcium response both post cue delivery ($^{**}p = 0.003$) and post TRPA1 stimulus ($^{***}p = 2.0 \times 10^{-5}$) was significantly lower in the presence of kin water as compared with water flow. For class $ii_{a-b}$, the integrated calcium response post cue delivery ($^{**}p = 0.0013$ or $^{***}p = 7.6 \times 10^{-7}$) was significantly lower with kin water as compared with water flow, but the response to TRPA1 stimulus was not significantly lower ($p = 1$ or $p = 0.06$, respectively), two-sided Wilcoxon signed-rank test, $n = 16/126/40/47/109/64/64/34$ neurons (ia, ib, ic, iia, iib, iii, iv, and v, respectively). Source data are provided as a Source Data file.

protocol was performed on fish maintained either in isolation or in the presence of conspecifics (Fig. 7a). Subsequently, animals were presented with a large excess of fluorescently labeled paramecia, which reveals food ingestion on the basis of gut fluorescence quantitation after a short feeding period. We found that food-deprived fish maintained in isolation consumed significantly less paramecia than those maintained in groups of three animals (Fig. 7b). When group size was varied from 2 to 5 individuals, the food intake per animal scaled with group size (Fig. 7b). Water-borne cues from kin, but not visual contact with kin larvae, were sufficient to rescue the isolation-induced suppression of food intake (Fig. 7c).

Given that social isolation increases neuronal activity in a subset of OXT-positive neurons, we asked if OXT neurons are required for the social control of appetite. To this end, OXT neurons were chemogenetically ablated in larvae via OXT neuron-specific expression of bacterial nitroreductase (NTR)[44]. Animals were incubated with a group of conspecifics along with the prodrug metronidazole (MTZ), which results in the death of NTR-expressing cells, during the period from 5 to 7 dpf. This treatment resulted in loss of ~80% of NTR-expressing preoptic OXT neurons. At 8 dpf, these larvae, or MTZ-treated control animals lacking the NTR gene, were placed in either into small groups or isolated and then assayed for feeding as described above. We found that OXT neuron ablation enhanced food intake in isolated animals, while the food intake of fish maintained in groups was unchanged (Fig. 7d). Consistently, addition of an OXT receptor antagonist strongly increased food intake in isolated fish (Fig. 7e), whereas OXT agonists strongly suppressed food intake of fish kept in a group (Fig. 7f). These results indicate that OXT signaling is both necessary and sufficient to mediate the social modulation of food intake in larval zebrafish.

Last, we compared food intake in OXT homozygous null mutant animals with their heterozygous siblings. As expected, we found that null mutant animals did not modulate their food intake on the basis of social environment, whereas the behavior of

their heterozygous siblings was indistinguishable from that of wild-type animals. However, mutant animals had low food intake in both the isolated and the group setting (Fig. 7g). Thus, as with acute ablation of OXT neurons, social modulation of feeding was impaired. However, the generally low food intake in these fish might reflect long-term deficits in circuit activity that altered the development or maintenance of neural circuits involved in socially modulated behaviors. In summary, these results demonstrate that an olfactory pathway, including OXT neurons, modulates diverse social behaviors in larval zebrafish.

## Discussion
Until recently, larval zebrafish had not been thought to exhibit robust social interactions, aside from simple behaviors such as avoidance of other larvae[12–14]. An emerging body of literature however has revealed the profound influence of visual, chemosensory, and mechanosensory social cues on larval zebrafish development and behavior[12,15,16,34,45]. Some studies have reported early kin odor preference and imprinting, suggesting an early experience-dependent development of positive social behaviors[16,18]. Here, we demonstrate that the presence of conspecific fish and chemosensory conspecific cues significantly increases appetite and reduces nociceptive responses in larvae, lending further support for the existence of early prosocial behaviors in zebrafish.

We further demonstrate that at these larval stages, even brief (~2 hr) social isolation results in distinct neural signatures that include aversive and nociceptive[6,7] brain circuits in the preoptic area, posterior tuberculum, caudal hypothalamus, and, notably, populations of oxytocin (OXT)-expressing neurons. These results are consistent with those of a recent study[45] that described increased activity in the preoptic area as a result of long-term social isolation in older (juvenile, 21 dpf) zebrafish. Thus, in an intriguing parallel to humans[46], we observe that pain and social isolation exhibit a shared neural signature in the larval zebrafish brain that extends also to older animals of the species.

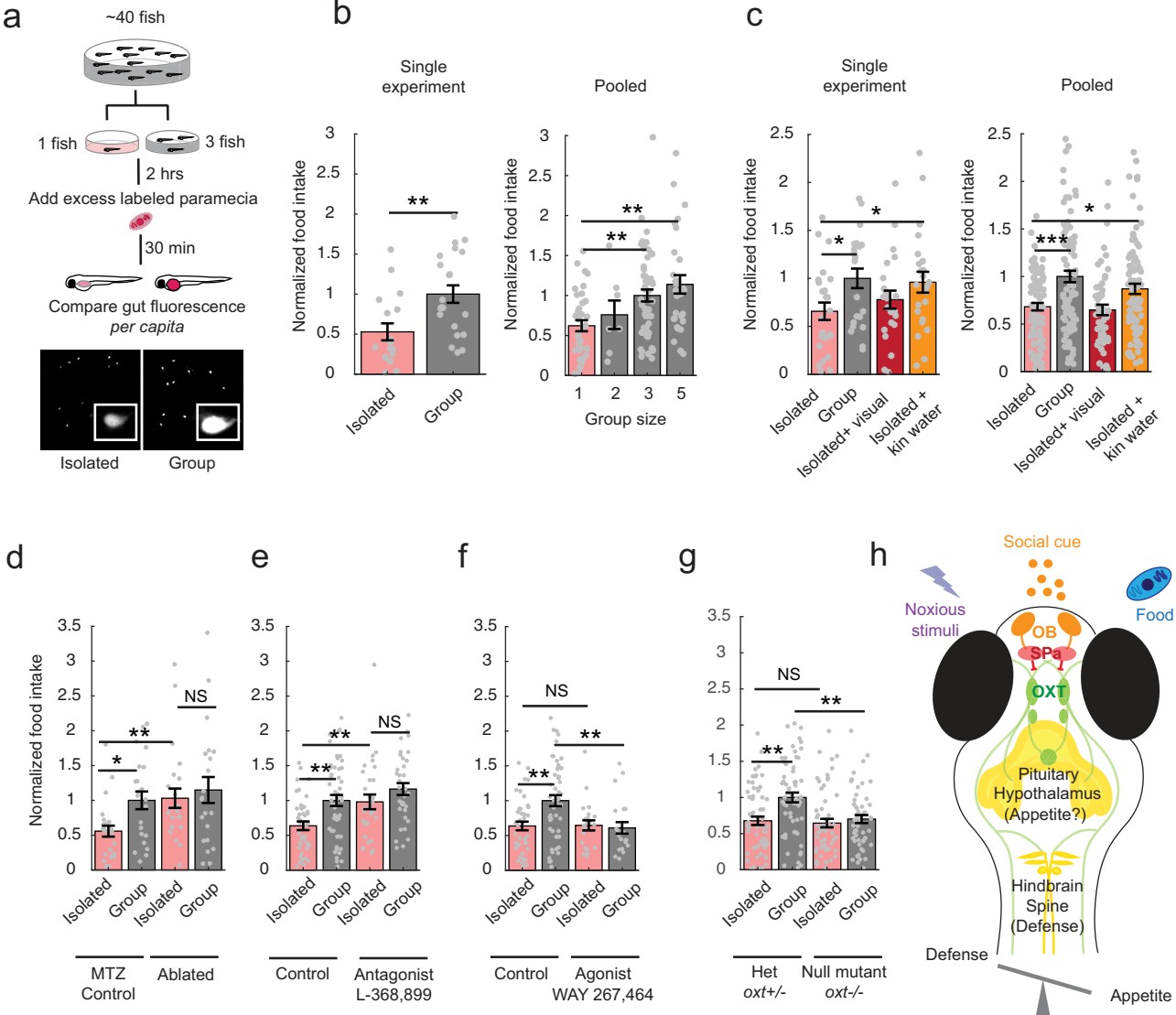

Our analysis of the OXT-expressing population reveals that these neurons, resembling a "hub and spoke" model[47], are functionally diverse and modulate distinct behavioral outputs, including feeding and nocifensive escape responses (see also Wee et al., 2019[6]). Further, we show that OXT neurons receive social information at least in part via an olfactory bulb to the subpallial circuit. A number of studies have implicated cell populations of the anterior subpallium (aSPa) in distinct visual and olfactory social responses[17,21,34,36]. Our analysis of the GABAergic y321:Gal4 population provides convergent evidence that the zebrafish subpallium, thought to be homologous to the mammalian medial amygdala[17,18,21], is a multimodal, social-cue-responsive, inhibitory circuit that regulates OXT activity.

We thus propose a circuit mechanism by which the larval zebrafish integrates conspecific social cues into the modulation of defensive and appetitive behaviors (Fig. 7h). Specifically, our data suggest that OXT neurons and other preoptic populations encode a range of chemical social information at least partially via olfactory inputs that then modulate nocifensive and appetitive behavioral outputs via downstream projections to the hindbrain and other hypothalamic regions. The role of other preoptic populations cannot be overstated—a recent study[7] confirmed that numerous preoptic neuropeptidergic populations including OXT are responsive to aversive stimuli, and that only co-ablation of OXT together with

corticotropin-releasing factor (CRF)-expressing neurons was sufficient to significantly attenuate aversive responses.

OXT neurons displayed a variety of responses to chemical social cues, which could uniquely signal the presence of adult fish (known predators), genetically related animals (larval kin), or distantly related (larval non-kin) conspecifics. Further, water flow itself elicited a mild response that may represent an arousing or mildly aversive cue. Overall, kin cues induced the most widespread inhibition of OXT neurons, while the same OXT neurons can be either excited or inhibited by other cues, suggesting diverse and valence-specific tuning. The significant heterogeneity in zebrafish OXT neuron activity and responses complements a larger body of both mammalian and fish literature describing heterogeneity in gene expression, projection patterns, receptor localization, and function of OXT neurons[8,48]. We hypothesize that activated OXT neurons could drive anxiolytic or antinociceptive effects, consistent with traditional views of OXT function[3].

We also note that while differences between kin and non-kin responses are clear in the olfactory bulb and subpallium, they are more subtle in $OXT_{PO}$ and surrounding preoptic neurons, and further, we have not demonstrated that such differential modulation results in distinct behavioral outcomes. However, they are consistent with a larger body of studies, which have reported the capacity for kin recognition and discrimination in zebrafish larvae

**Fig. 7 Manipulation of OXT signaling affects appetite in a social-context- dependent manner. a** Schematic depiction of social feeding behavior experiments. Example images are shown (inset shows higher (100x) magnification). **b** Left: Gut fluorescence in a single experiment demonstrates that isolated fish, on average, ingest less paramecia than fish maintained in a group of 3 conspecifics. All gut fluorescence measurements are normalized to the group mean (**$p = 0.0058$, $n = 21/17$ fish, two-sided Wilcoxon rank-sum test). Right: Normalized food intake (gut fluorescence normalized to the average for groups of 3 fish) scales with group size (1 fish, $n = 36$; 2 fish, $n = 7$; 3 fish, $n = 52$; 5 fish, $n = 29$ over 4 experiments). **$p = 0.0065$ (1 fish vs 3 fish)/**$p = 0.0035$ (1 fish vs 5 fish), two-sided Kruskal–Wallis test with correction for multiple comparisons. Data are presented as mean values ±SEM. **c** Water-borne but not visual cues rescue reduced feeding of isolated fish. Left: Results within a single experiment. *$p = 0.03$ (isolated vs group), *$p = 0.04$ (isolated vs kin water), $p = 0.17$ (isolated vs visual), $n = 24/24/22/23$ fish, two-sided Wilcoxon rank-sum test. Right: Average of 6 experiments, $p = $ ***$2.8283 \times 10^{-4}$ (single vs group), *$p = 0.02$ (single vs kin water), $p = 0.35$ (single vs visual), $n = 86/86/55/87$ fish, two-sided Wilcoxon rank-sum test. Data are presented as mean values ±SEM. **d** Cell-specific chemogenetic ablation of OXT neurons specifically rescues the effect of social isolation on appetite. OXT neurons were ablated via OXT neuron-specific expression of bacterial nitroreductase using *Tg(oxt:Gal4;UAS:nfsb-mCherry)* fish[6, 44]. Animals were incubated with a group of conspecifics and the prodrug metronidazole (MTZ) during the period from 5 to 7 dpf. This treatment resulted in loss of ~80% of nitroreductase-labeled preoptic OXT cells (unablated = $20.3 \pm 1.1$ neurons, ablated = $4.5 \pm 0.5$ neurons; $n = 20$ control fish, 29 fish with ablation). *$p = 0.015$ (isolated control vs group control), **$p = 0.0017$ (isolated control vs single ablated), $p = 0.76$ (isolated ablated vs group ablated), $n = 65/59/57/56$ fish over 5 experiments, two-sided Wilcoxon rank-sum test. Controls are metronidazole (MTZ)-treated, non-transgene-expressing siblings. Data are presented as mean values ±SEM. **e** The oxytocin receptor antagonist (L-368,899) restores social feeding levels to isolated animals. Antagonist (5uM) was added to the incubation water at the start of the 2-hr isolation period (see (**a**)). Data are presented as mean values ±SEM. **$p = 0.0022$ (single vs group), **$p = 0.0091$ (single control vs single antagonist), $p = 0.09$ (single antagonist vs group antagonist), $n = 40/50/29/34$ fish over 3 experiments, two-sided Wilcoxon rank-sum test. Data are presented as mean values ±SEM. **f** An oxytocin receptor agonist (WAY 267,484) ($5\,\mu M$) reduces group food ingestion to the level of isolated animals. Agonist was added to the incubation water at the start of the 2-hr isolation period (see (**a**)). **$p = 0.0022$ (single vs group), $p = 1$ (single control vs single agonist), **$p = 0.0069$ (group control vs group agonist), $n = 40/50/20/21$ fish over 3 experiments, two-sided Wilcoxon rank-sum test. Control groups for both (**e**) and (**f**) are the same sets of fish, split up for better visualization. Data are presented as mean values ±SEM. **g** Comparison of food intake of *oxt* null mutants (*oxt−/−*) and their heterozygous wild-type siblings (*oxt+/−*) in isolated and nonisolated contexts Single-group differences in food consumption are abolished—however, food intake was reduced in groups, rather than enhanced in isolated fish. **$p = 0.0011$ (single vs group), $p = 0.65$ (single control vs single mutant), **$p = 0.0012$ (group control vs group mutant), $n = 60/59/58/54$ fish over 5 experiments, two-sided Wilcoxon rank-sum test. Data are presented as mean values ±SEM. **h** Model for how oxytocin neurons integrate information on social state to control appetite and nocifensive behaviors. We posit that social chemical cues are olfactory, and that GABAergic neurons in the subpallium transform olfactory bulb activation into inhibitory signals that differentially modulate the OXT population. The OXT circuit modulates nocifensive behavior via brainstem premotor neurons[6]. The OXT neurons project extensively to other areas of the hypothalamus and the pituitary; these downstream regions may be involved in mediating effects on appetite[43]. Source data are provided as a Source Data file.

of comparable ages[16,18]. Further, our results raise the likelihood that OXT neurons in the PO and PT may respond differentially to social cues. For example, our pERK results (Fig. 1) suggest that $OXT_{PT}$ neurons may be more strongly suppressed by non-kin and visual rather than kin cues. Calcium imaging (Fig. 2) also revealed differences in $OXT_{PO}$ and $OXT_{PT}$ activities, though the low numbers of $OXT_{PT}$ neurons recorded make it difficult to identify conclusive differences. Finally, y321 optogenetic stimulation was sufficient to suppress $OXT_{PO}$ but not $OXT_{PT}$ neuron activity (Fig. 5), again suggesting that $OXT_{PO}$ and $OXT_{PT}$ may differentially encode or respond to social stimuli. Intriguingly, a previous study[23] had demonstrated a specific role of $OXT_{PT}$ but not $OXT_{PO}$ neurons in the development of social preference, again suggesting differences in their functional roles.

In our study, we did not attempt to identify the chemical components within kin or conspecific-conditioned water that are sufficient to modulate OXT activity and behavior, though it is an interesting avenue for follow-up work. Previous studies have suggested the role of bile acids[49,50] or MHC peptides[20] as odorants that mediate social or kin recognition. Our clustering analyses have demonstrated complexity in the encoding of kin, non-kin, and adult water stimuli by olfactory and subpallial neurons, which strongly suggests that multiple components, rather than a single compound, are contributing to kin versus conspecific discrimination.

In humans and other mammals, it is known that social cues, including odors, can attenuate aversive experience and behaviors, a phenomenon known as "social buffering"[38,39]. Social facilitation of appetite has also been observed in many species[40], and is likely evolutionarily adaptive. In general, an isolated animal needs to shift its priorities from foraging to vigilance or escape, since it may be more susceptible to the risk of predation. Accordingly, adult zebrafish display isolation stress in a group size-dependent

manner[51]. We had previously shown that OXT neurons respond to a range of aversive, particularly noxious stimuli, and are sufficient to drive motor responses by acting on brainstem targets[6]. We also previously showed that ablation of OXT neurons or OXT gene deletion had a moderate but significant effect on nocifensive behavior[6]. By demonstrating that chemical social cues converge on this circuit, and that kin cues, in particular, predominantly diminish the activity of TRPA1-responsive OXT neurons, we provide a potential mechanistic understanding of how the OXT circuit can mediate the phenomenon of "social buffering" in a vertebrate organism[52].

Although we have demonstrated an attenuative effect of kin cues on behavior, including nocifensive responses, we note that we have not demonstrated the converse—that isolation can enhance aversive behavioral output. More detailed analyses of nocifensive behavior in isolated versus grouped larvae, or of the effect of isolation duration on nocifensive behavior, will help bolster our hypothesis. We also note that the genetic and pharmacological manipulation experiments of OXT signaling on food uptake (Fig. 7) shown in this paper cannot be generalized to its social buffering effects, and that additional circuit manipulation experiments will be necessary to directly tie OXT to social-cue-dependent attenuation of nociceptive responses. Further, given the overlapping sensory and behavioral effects of other neuropeptides in the preoptic area[7], it is likely that OXT is not the sole mediator of social buffering behavior, and that manipulation of other preoptic populations (e.g., CRF-expressing neurons) would be necessary to completely abolish social-dependent influences on behavior.

Notably, the attenuative effect of OXT signaling on larval zebrafish feeding behavior is consistent with mammalian studies: (1) the insatiable appetite and morbid obesity observed in Prader–Willi syndrome is likely due to impaired OXT

signaling[53,54]; (2) acute inhibition of paraventricular OXT neurons can promote food intake[55]; (3) lesions of the PVN, as well as mutations that affected OXT neuron development, have been shown to cause hyperphagia and obesity[56,57]; and (4) direct administration of OXT has been shown to reduce feeding[4,5]. At the same time, our data suggest that the role of OXT in feeding may be more complex than previously appreciated. Notably, while OXT homozygous mutants do not show social state-dependent modulation of appetite, they also eat less than their heterozygous siblings, implying that a complete absence of OXT may be detrimental toward feeding.

Such a profound influence of social context on OXT's appetite-suppressing effects may very well generalize to mammals. For example, a recent study found that inhibiting OXT signaling enhances sugar intake in a socially dominant mouse, regardless of their social context, whereas in subordinate mice, such inhibition only enhanced appetite when cues from the dominant mouse were not present[58]. Our results also complement the observations of strong interactions between social and feeding circuits across evolution[59,60], and reinforce a role for OXT in prioritizing different motivated behaviors[4]. However, the effects of OXT in larval zebrafish may occur as part of a coordinated response to both social isolation and noxious contexts, rather than reproduction or parental care.

There are important distinctions between our findings and the canonical view of OXT function as suggested by mammalian studies, the most significant of which is that the larval zebrafish OXT neurons show widespread *activation* by social isolation, rather than by conspecific cues[2,61], and may in fact represent a negative valence state in larval zebrafish, rather than the rewarding experience it is generally associated with.

We propose three possible reasons for these apparent differences: *first*, representations within both the mammalian and zebrafish OXT population are diverse, and thus the observed activity patterns in zebrafish may reflect those of specific sub-populations of mammalian OXT cells (indeed, some zebrafish OXT neurons are activated by conspecific cues, and some mammalian OXT cells are inhibited by conspecific cues[61]). Interestingly, recent studies in mammalian models have demonstrated that OXT neurons can also be negatively reinforcing, and promote fear, stress, and anxiety in some situations[1,62]. *Second*, OXT response properties may have changed over the course of evolution, as more sophisticated social functions were derived. *Third*, there is a possibility that OXT neuron response properties might reverse over the course of development, since adult and kin odor generate opposite activity signatures; however, since the enhancement of preoptic area activation appears to persist at least till juvenile stages[45], when social preference behaviors have developed, any such reversal would have to happen closer to adulthood. Intriguingly, a recent study[63] has demonstrated that OXT receptor loss-of-function in fact enhances social development in zebrafish, confirming our findings that OXT signaling in zebrafish may not necessarily be prosocial. Overall, our results may provide a broader and more intricate perspective of this highly conserved peptide's social function and evolution in vertebrate animals.

In conclusion, our study demonstrates how organizing principles and circuit-implementation strategies underlying social behavior can be elucidated by probing social context-dependent behaviors in a small and optically accessible model organism. More broadly, our dissection of the larval zebrafish OXT circuit provides an entrypoint into understanding how neuromodulatory systems represent behavioral states such as social isolation, hunger, and acute nociception, on multiple timescales, and how these representations are then used to modulate behavioral output in a flexible and context-dependent manner.

## Methods

**Fish husbandry and transgenic lines**. Adults were raised in facility water, and larvae in embryo water or filtered facility water, and maintained on a 14:10-h light:dark cycle at 28 °C. All protocols and procedures involving zebrafish were approved by the Harvard University/Faculty of Arts & Sciences Standing Committee on the Use of Animals in Research and Teaching (IACUC). Fish were raised at a density of ~40 fish per dish and fed paramecia from 5 dpf till the day of the experiment. Behavioral experiments were carried out mostly on fish of the WIK background, although other genotypes (e.g., AB, or mit1fa−/− (nacre) in the AB background) were also utilized and showed similar behavioral results. mit1fa−/− (nacre) in the AB background, along with additional transgenes described below, were also used for calcium imaging and MAP-mapping experiments. Transgenic lines Tg(oxt:GFP)[64], Tg(HuC:GCaMP6s)[65], Tg(UAS:GCaMP6s)[66], Tg(UAS:nfsb-Cherry)[44] [nitroreductase is referred to as NTR in the text], Tg(UAS:ReaChR-RFP)[37,43], Tg(Gad1b:loxP-dsRed-loxP-GFP) [referred to as Tg(Gad1b:RFP) in text][67], Tg(y321:Gal4)[34,68], Tg(oxt:Gal4), and oxytocin mutants[6] were previously published.

**MAP-mapping**. In total, 7–8-dpf larvae, that had been continuously fed with an excess of paramecia since 5 dpf, were either isolated or split into small groups, using 35-mm dishes filled with 3 ml of embryo water. For groups of 10, a larger (10-cm) dish was used to prevent overcrowding. Paramecia was present within each dish to ensure that the fish were well-fed and had ample stimulation. After 2 hrs, larvae were quickly funneled through a sieve, which was then quickly dropped into ice-cold 4% paraformaldehyde, immunostained (see next section), mounted dorsal-side up in agarose on a petri dish, imaged using multi-positioning software on an Olympus FV1000, and analyzed using publicly available code as reported in Randlett et al. (2015)[22]. FDR threshold used = 0.05%.

**Whole-mount immunostaining**. About 24 hrs after fixation (4% paraformaldehyde (PFA) in PBS), fish were washed in PBS + 0.25% Triton (PBT), incubated in 150 mM Tris-HCl at pH 9 for 15 min at 70 °C (antigen retrieval), washed in PBT, permeabilized in 0.05% Trypsin-EDTA for 45 min on ice, washed in PBT, blocked in blocking solution (10% goat serum, 0.3% Triton in Balanced Salt Solution, or 2% BSA in PBS, 0.3% Triton) for at least an hour and then incubated in primary (rabbit anti-pERK, Cell Signaling #4370 and/or mouse anti-ERK (tERK), Cell Signaling #4696, 1:500) and AlexaFluor secondary antibodies (1:500) for up to 3 days at 4 °C diluted in blocking solution. In-between primary and secondary antibodies, fish were washed in PBT and blocked for an hour. If necessary, pigmented embryos were bleached for 5 min after fixation with a 5%KOH/3%$H_2O_2$ solution.

The protocol was similar for dissected brains, except that the brains were dissected in PBS after 24 h of fixation, and the permeabilization step in Trypsin-EDTA and Tris-HCL antigen retrieval was omitted. Dissected brains were mounted ventral toward the coverslip on slides in 70% glycerol prior to imaging. Confocal images of dissected brains were obtained using either a Zeiss LSM 700 or Olympus FV1000.

**Exposure to sensory cues (for pERK experiments)**. For generation of conspecific-conditioned water, sibling or nonsibling larvae that had been continuously fed with an excess of paramecia from 5 dpf were transferred into a new 10-cm petri dish that did not contain any paramecia, at a concentration of 2 fish per ml. After a 2-hr incubation, a syringe with an attached 0.45-μm filter was used to very gently draw out the conditioned water, with great care taken not to disturb or stress the fish in the process.

In total, 7–8-dpf larvae, that had been continuously fed with an excess of paramecia since 5 dpf, were either isolated or split into small groups, using 35-mm dishes filled with 3 ml of embryo water. Paramecia was present within each dish to ensure that the fish were well-fed and had ample stimulation. About 700 μl of the filtered conspecific-conditioned water was added to each 35-mm dish (~1:5 dilution), 30 min before fixation, and embryo water was correspondingly added to controls. For providing visual access to conspecifics, the 35-mm dishes containing single larvae were inserted into larger (55 mm) dishes containing ~5 larvae that are surrounding but unable to interact with the single larva. After 2 hrs, larvae were quickly funneled through a sieve, which was then quickly dropped into 4% paraformaldehyde, dissected in PBS, and immunostained as described above and in Wee et al., 2019[6].

**High-resolution pERK analysis**. For quantification of pERK intensities of individual OXT neurons, pERK experiments were performed on dissected Tg(oxt:GFP) brains. Cellular-resolution imaging of dissected brains was obtained using the Zeiss (LSM 700 and LSM 880) or Olympus (FV1000) confocal microscopes. pERK intensities of individual GFP-positive (OXT) or GFP-negative (non-OXT) neurons were measured using Fiji/ImageJ[69] and quantified using MATLAB as also reported in Wee et al., 2019[6] (codes are publicly available). In Supplementary Fig. 1a, b, pERK intensities shown are normalized to the respective mean PO or PT intensities of control fish (i.e., fish maintained in groups of 3), whereas in Fig. 1c and Supplementary Fig. 1c, pERK intensities are normalized to the mean of both PO and

PT intensities of control fish. PO and PT were segmented based on their anterior–posterior positions.

For quantification of pERK intensities of y321 neurons (Supplementary Fig. 6), GFP-positive cells were manually segmented. Raincloud plots[70,71] were produced using open-source software (version 2) in Fig. 1 and Supplementary Fig. 1. For sampling of neurons for spatial distribution analysis, a fixed number of neurons were first randomly sampled without replacement per fish, and then another round of random sampling was performed to ensure even distribution across treatment groups (as different treatment groups had different numbers of fish). This ensured a more even representation of neurons across all fish and treatments. "Active cells" are cells that have normalized pERK values surpassing the threshold by which only 10% of control (PO and PT combined) cells would be considered active.

**Calcium imaging and olfactory stimulation**. In all, 8–11-dpf larval *Tg(oxt:Gal4;UAS:GCaMP6s)* fish in the nacre background were used for calcium imaging experiments. They were embedded in the center of a 55-mm dish in 1.5% agarose with their tails and noses freed. Fish were isolated for between 30 min and 4 h prior to the experiment, with the bulk between 1 and 3 h. For volumetric calcium imaging, an electrically tunable lens (Edmund Optics, 83-922) was installed in the light path before the galvanometer scanner to allow fast axial refocusing of the two-photon excitation spot. Images alternating between two or three z-planes per fish were acquired at a frame rate of ~237 ms per z-plane, corresponding to ~474 ms or ~711 ms per z volume, respectively.

Kin or non-kin-conditioned water (at a concentration of 1 fish/ml) was generated as described above. In these calcium imaging experiments, we used WIK or EK fish as non-kin fish, since the *Tg(oxt:Gal4;UAS:GCaMP6s)* fish we imaged were of the AB genetic background. For adult water, 5 adult kin (from the parent tank of larvae) were used to condition 500 ml of water (1 adult fish/100 ml) for 2 hrs, and the water also subsequently filtered. Kin and non-kin fish used to condition water were raised apart from experimental fish from 3 dpf, to dissociate genetic from familiarity effects.

Alternating water or chemical stimuli were delivered using a custom-built syringe pump system controlled by custom Labview software. At specified time intervals, 300 µl of each cue (~30 µl/second) was delivered using a zero-dead-volume multichannel perfusion pencil (AutoMate Scientific). Embryo water was also constantly circulated through the dish using a peristaltic pump (Harvard Apparatus). The interstimulus interval (ISI) was 2 min for olfactory bulb imaging and 5 min for OXT neuron imaging. The longer ISI for OXT neuron imaging was implemented to reduce desensitization and ensure that activity truly returned to baseline before presenting the subsequent stimulus. In order to reduce experiment time and avert the possibility of OXT neuron desensitization, we also never compared responses to more than 2 cues (e.g., either kin vs non-kin, or kin vs adult, but not all 3 stimuli).

For *Tg(y321:Gal4;UAS:GCaMP6s)* imaging experiments, solenoid-controlled (Automate Scientific) gravity flow[72] (Herrera et al. 2021) was used to deliver the four stimuli (control-filtered facility water, kin, non-kin, and adult water) to each fish. The stimulus delivery cycle was as follows: stimulus was actively flowed onto the fish for 20 s, followed by 45 s of no flow, which was followed by 20 s of flow of control water to wash out the chemical, and finally another 45 s of no flow.

Calcium imaging and behavioral monitoring with TRPA1 stimulation was performed as previously reported[6] on 8–10-dpf larvae, with a number of core differences: (1) 15 µM instead of 25 µM optovin was used, to reduce background signals (see below); (2) nostrils are exposed to allow for olfactory stimulation; and (3) the same UV stimulus intensity used throughout the experiment, and only 4 stimulations at 5-min intervals were presented (alternating water and kin stimulus, equal number of fish with the starting with kin or water stimulus).

We raised larvae and performed all experiments in embryo water maintained at pH 6.5–7.5, which is crucial for ensuring optimal behavioral responses to TRPA1 stimulation. For TRPA1 photoactivation, fish were incubated in optovin (Tocris Biosciences 4901, 15 µM in 0.1% DMSO, vortexed and filtered with a 0.45 µm syringe filter to remove undissolved particles) for 5–10 min. An ultraviolet laser was focused on the middle of the tail (7 mW at sample) and aligned on non-experimental fish before starting the experiment to prevent unnecessary exposure in the experimental animals. Long (seconds to minutes). An electronic shutter (Uniblitz VS25) was installed above the objective to prevent saturation of the photomultiplier tube by the ultraviolet laser. Tail movements were simultaneously monitored under infrared (850 nm) illumination at 200 fps (Pike F-032 camera, Allied Vision Technology) through a substage optical path, and tail coordinates were extracted online using custom LabVIEW software.

Fish were isolated for between 1.5 and 6 h prior to the experiment, with the bulk between 2 and 3 h. Since, in addition to the tail, the nose was exposed in this current paradigm, Optovin, a colored solution, was rapidly absorbed into the fish's brain and caused a linear increase in background signal (i.e., even in non-GCaMP-labeled tissue) over time. Post hoc subtraction of this background signal from OXT GCaMP signals restored a flat baseline, allowing us to extract meaningful calcium signals.

**Data analysis for calcium imaging**. All calcium imaging data were analyzed using custom Fiji and MATLAB software. The general protocol for analysis was:

1. Image registration to correct for motion artifacts using the TurboReg[73] plugin in Fiji/ImageJ.
2. Extraction of fluorescence signals from both channels using manually or correlation-based (see code in Herrera et al. 2021[72]) segmented ROIs in MATLAB or Suite2p[31]. For the analysis of *Tg(y321:Gal4;UAS:GCaMP6s)* neurons, manually drawn ROIs were created around the cell body regions of the anterior subpallium to isolate automatically segmented ROIs that were from soma. For fish expressing *Tg(HuC:GCaMP6s)*, cell segmentation from correlated voxels was performed and curated using Suite2p software. RFP or mCherry-expressing cells were classified as having a probability threshold of >0.4 (Supplementary Figs. 4, 5).
3. Calculation of Δf/f signals from raw traces and alignment to tail traces, as needed, in MATLAB. Δf/f values were calculated from raw traces using the average fluorescence over the time period before the first stimulus as the baseline to which all traces were normalized. Behavior (if any), stimulation, and calcium imaging timestamps were aligned and used to extract stimulus-triggered averages as well as to generate motor and /or stimulus regressors to correlate with calcium activity. The regressors were convolved with a GCaMP6s kernel based on its measured response delay (0.48 s) and decay time (3 s) and cross-correlated with calcium traces that had been smoothed with a 3-frame zero-phase filter. Bayesian information criterion and Aikake information criterion analysis was used to determine optimal number of clusters for K-means clustering of olfactory bulb neuron correlation coefficients with different regressors, and the same number of clusters was eventually adopted across all datasets for simplicity.

To determine if an OXT neuron was activated or suppressed, we averaged the calcium signal over a 60-s interval post stimulus for each stimulus type. In the case of simultaneous TRPA1 stimulation, integrated activity was quantified either 5 s post TRPA1 stimulation, or 30 s prior to TRPA1 stimulation (i.e., post cue delivery but pre TRPA1 stimulation). If the difference between the two integrated calcium signals during olfactory and water stimulation was more than 0.05, we classified the neurons as being either activated or suppressed by the cue, depending on the sign. Although the threshold of 0.05 is arbitrary, we show that across a range of thresholds, the relationship between the proportions of suppressed and enhanced neurons remains consistent for each type of stimulus (Supplementary Figs. 2c, 8d). For accurate comparison of OXT neuron activity and spatial distributions in Fig. 2, we randomly sampled 25 neurons per fish, then 200 neurons were randomly selected from each group.

For tethered behavior analysis, bouts were first extracted using the 'findpks' function in MATLAB. The number of large or small-angle bouts within 90 s prior to cue delivery (precue period), 30 s from cue delivery (post cue period), or 5 s of TRPA1 stimulus were subsequently quantified. Other parameters quantified included maximum and minimum velocities and peak cumulative tail angles (and latencies where applicable) of bouts occurring prior to or directly after TRPA1 stimulation.

**Calcium imaging and optogenetics (Fig 5)**. Optogenetic stimulation and calcium imaging was performed on a commercial 2-P microscope with confocal settings (Zeiss LSM 980 NLO Multiphoton) using a 633 nm laser for ReAChR activation, and a 488 nm laser for calcium imaging, at a frame rate of about 1 fps. *Tg(y321:Gal4; oxt:Gal4; UAS:ReAChR-RFP; UAS:GCaMP6s)* transgenic fish were used to record OXT activity after y321 ReAChR activation. Fish were embedded in 1.8–2% agarose. ReAChR activation was induced in the telencephalon periodically for 10 s, and ensuing activity in the preoptic area/posterior tuberculum was recorded continuously during the 60–180-s interval between stimuli (Table 1). The averaged response of each y321 or OXT neuron to light stimuli was integrated across a 40-s post-stimulus time window for statistical comparison between treatments, with this epoch split into two (first 10 s, subsequent 30 s) for OXT neuropil analysis.

**Free-swimming TRPA1 stimulation (Supplementary Fig. 7)**. Fish were singly placed into a 20.6-mm cut-out agarose circular mold illuminated by three quad blue LEDs (Luxeon Star, 470 nm). To probe the effect of conspecific cues, DMSO or Optovin solutions were generated either in embryo water or kin-conditioned water. Following 5 min of habituation in DMSO (effectively also the isolation period), fish were stimulated once every 30 s with a 100-ms pulse of blue light. The DMSO solution was then exchanged with Optovin, and the same protocol repeated. Behavior was recorded at 200 fps (Pike F-032, Allied Vision) and analyzed using custom Python software.

**Social feeding experiments**. For experiments in which feeding was assessed, larvae that had been continuously fed with an excess of paramecia from 5 dpf were either isolated or placed in groups of 3, in 35-mm dishes (3 ml of embryo water), in the absence of food. Paramecia was labeled using lipid dye (DiD' solid, D-7757, Thermo Fisher Scientific, dissolved in ethanol) for ~2 hrs (5 µl of 2.5 mg/ml working solution per 1 ml of concentrated paramecia, followed by spinning down, rinsing, and reconstitution in deionized or embryo water). After 2 hrs of food deprivation, fluorescent-labeled paramecia was added followed by a quick fixation after 30 min (full protocol is described in Wee et al. (2019)[43] and available here:

## Table 1 Summary of optogenetic experiments performed.

| | y321 ReaChR (record y321) | y321 no ReaChR (record y321) | y321, oxt ReaChR (record OXT) | Negative control |
|---|---|---|---|---|
| 1 | *7 stims/180 s | 10 stims, 60 s | *11 stims/180 s | 10 stims/180 s |
| 2 | *6 stims/60 s | 14 stims, 60 s | *10 stims/180 s | 7 stims/120 s |
| 3 | *6 stims/120 s | 12 stims, 60 s | *13 stims/180 s | 12 stims/60 s |
| 4 | *7 stims/120 s | 14 stims, 60 s | *13 stims/120 s | 13 stims/60 s |
| 5 | *7 stims/120 s | | *13 stims/120 s | 12 stims/60 s |
| 6 | *7 stims/120 s | | *13 stims/120 s | |
| 7 | *9 stims/60 s | | *14 stims/60 s | |
| 8 | *8 stims/60 s | | *14 stims/60 s | |
| 9 | *8 stims/60 s | | *13 stims/60 s | |
| 10 | *7 stims/60 s | | *14 stims/60 s | |
| 11 | *7 stims/60 s | | *13 stims/60 s | |
| 12 | *7 stims/60 s | | *14 stims/60 s | |
| 13 | *6 stims/60 s | | *12 stims/60 s | |
| 14 | 6 stims/60 s | | 8 stims/60 s | |
| 15 | 10 stims/60 s | | 7 stims/60 s | |
| 16 | | | 12 stims/60 s | |
| 17 | | | 10 stims/60 s | |
| 18 | | | 10 stims/60 s | |

Summary of experiments performed, stating both the number of stimulations (stims) and the interstimulus interval (in seconds). Asterisks indicates that the same fish was tested in under both protocols.

https://bio-protocol.org/prep1116). Sensory cues were generated and presented as described above for pERK experiments. Fixed larvae were subsequently distributed into 96-well flat-bottom dishes and imaged using the AxioZoom V16 (Zeiss) and analyzed using Fiji software (3D Objects Counter, custom software also available on www.github.com/carolinewee).

**Pharmacological experiments**. For pharmacology experiments, the OXT antagonist L-368,899 hydrochloride (Tocris Biosciences, 2641, 5 μM) and agonist WAY 267464 dihydrochloride (Tocris Biosciences, 3933, 5 μM) were added to each dish from the beginning of the isolation period (2 hrs prior to adding fluorescently labeled paramecia).

**OXT neuron ablation/inhibition experiments**. Transgenic larvae expressing *Tg(oxt:Gal4;UAS:nfsb-mCherry)* or their non-transgenic siblings were incubated in 2.5 mM Metronidazole dissolved in embryo water (MTZ, Sigma-Aldrich, M3761) from 5 to 6 or from 6 to 7dpf. MTZ was subsequently washed out, and food intake was measured at 7 or 8 dpf, respectively. For these experiments, the MTZ-treated siblings were used as the control group.

**Oxytocin mutants**. For experiments with *oxt* mutants[6], a heterozygous *oxt* mutant was crossed to a null mutant. All fish from the same clutch were raised together in the same dish and fed from 5 dpf. On the day of the experiment (7 or 8 dpf), the larvae were distributed into isolated or nonisolated conditions and fed labeled paramecia after 2 hrs according to the social feeding protocol described above. They were fixed after 30 min, transferred into 96-well plates to keep track of experimental conditions, and their gut fluorescence was imaged. After imaging, their genomic DNA was extracted and genotyped, and their genotypes were matched post hoc to their experimental conditions (isolated or nonisolated).

**Statistics**. All error bars show mean ± SEM over fish. Significance was reported as follows: $*p < 0.05$, $**p < 0.01$, $***p < 0.001$. Significance was determined using the Wilcoxon signed-rank test for paired data and the Wilcoxon rank-sum test for independent samples. One-sided tests were used in cases where there was a clear hypothesis for the direction of effect. The Kruskal–Wallis test with Tukey–Kramer correction was used in cases where there were multiple comparisons. Wilcoxon signed-rank test was used in Fig. 1a for comparing the distribution of normalized OXT ROI signals across different behavioral stimuli to the null hypothesis of median 1.

**Reporting summary**. Further information on research design is available in the Nature Research Reporting Summary linked to this article.

## Data availability

Source data are provided with this paper. Any additional data and resources (transgenic lines/mutants generated) will be made available by the corresponding author upon request.

## Code availability

Live versions of the analysis code are maintained in the following Github repositories: https://github.com/carolinewee/CelullarpERKanalysis/v1.0.0, https://doi.org/10.5281/

zenodo.5886670[74], https://github.com/carolinewee/CalciumAnalysisMATLAB/v1.0.0, https://doi.org/10.5281/zenodo.5894669[75]. https://github.com/carolinewee/CalciumAnalysisFiji/v1.1.0, https://doi.org/10.5281/zenodo.5894736[76]. https://github.com/carolinewee/gutfluorescence/v1.0.0, https://doi.org/10.5281/zenodo.5894734[77]. MAP-mapping code is available on (http://engertlab.fas.harvard.edu/Z-Brain/).

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

## Acknowledgements

We are grateful to Adam Douglass (Univ. of Utah) for continued support, advice, and comments. We thank the CBS imaging facility, Harvard Center for Biological Imaging, and the NorthWest Undergraduate Teaching Laboratories at Harvard for the successful completion of many experiments. J. Miller, K. Hurley, and B. Hughes provided invaluable fish care. This work was supported by U01-NS090449 (FE), R24-NS086601 (FE), U19-NS104653 (FE and SK), and Simons Foundation grant SCGB 325207 (FE). CLW was supported by the National Science Scholarship from the Agency for Science, Technology and Research (A*STAR), Singapore.

## Author contributions

C.L.W., E.S., and S.K. originally conceived of the project, which was then developed into its final form with M.N and F.E. S.K. and F.E. supervised the project. C.L.W., E.S., and M.N. designed and performed most of the experiments, and analyzed most of the data. K.J.H. and E.S. performed y321 calcium imaging and optogenetic imaging experiments, K.J.H and C.L.W. analyzed data from these experiments. M.N. also developed hardware and software for calcium imaging and behavioral experiments, and analyzed the free-swimming behavioral data. S.W. performed feeding experiments and analyzed data. C.L.W., S.K., and F.E. wrote the paper with contribution from all other authors.

## Competing interests

The authors declare no competing interests.

## Additional information

**Peer-review information** *Nature Communications* thanks Yunlu Zhu, Emre Yaksi, and the other anonymous reviewer(s) for their contribution to the peer review of this work. Peer reviewer reports are available.

