## [Peer review file · Nature Communications]

REVIEWER COMMENTS

Reviewer #1 (Remarks to the Author):

Here, Wee et al presented a study investigating the role of OXT neurons in regulating animals' responses to appetitive and aversive cues, and how such regulation can be controlled by olfactory social cues. The authors first presented a behavioral assay where, they showed the alterations of brain activity in response to social isolation, using pERK method. The authors argue that a sub population of OXT neurons increase their activity in response to social isolation, and this increase can be dampened by non-kin and kin water from larval fish. Next, using calcium imaging the authors showed that olfactory regions of the zebrafish brain respond to non-kin/kin water from larval fish, and adult fish. Later the authors showed that a sub population of OXT neurons decrease their calcium activity, in response to non-kin/kin water from larval fish, and another population increase their calcium activity when presented with adult fish water. Next, the authors argue that the kin water from larval fish modulate OXT nociceptive responses. Later, the authors studied the effect of social isolation in feeding behaviors, and showed that animals raised in bigger groups eat more. Finally the authors showed that genetic/pharmacological silencing of OXT system alter feeding behavior in isolated animals but not animals raised in groups.

Overall, the behavioral part of this study is interesting and solid. However, the functional investigations of the phenomenon are rather weak. Especially the results section is not well presented, and most of the functional results are only weakly supported by the presented data and analysis. The figures are confusing and not coherent; hence I find myself reading the text/figures/legends again and again trying to understand how the presented data support authors comments. In general, most functional effects pointed by the neural data are weak and convoluted. Altogether, I find authors hypothesis interesting, but only weakly supported by the presented neural data, which is rather preliminary and not well presented.

Please find below my comments, which I hope will help the authors to improve their study.

- 1) Figures are very confusing, and authors use several different and often convoluted approaches to present same type of data in different figures. Also, the results are not well described in the text or the figures. I highly recommend the authors to spend some time in preparing better figures and better analysis.
- 2) The link to the olfactory system is weak. How the olfactory system is linked to OXT system? Is it even clear that the presented effect is due to odors? I could imagine that water samples from different groups might have differences in pH, oxygen levels or other factors associated with the presence of animals. What content of the fish water carries the social information, and how this information reaches to OXT neurons is a major lacking point.
- 3) The diverse effects on OXT neurons is interesting, but the underlying reasons are not investigated well enough. Why some OXT neurons increase and some decrease their activity in response to odors?
- 4) The modulation of OXT neurons responses to noxious stimuli by kin-water is very weak as evident by only few selected data points showing such modulation. Better experimental design is necessary to show that such modulation is significant and reproducible.
- 5) I am surprised to see that the authors did not investigate sufficiently the encoding of different water samples (kin, non-kin, adult, etc) in the olfactory or OXT neurons

Minor comments

- 1) Do the animals have social interactions at this age that is studied? Previous studies from several zebrafish labs suggest that social interactions are rather limited at these developmental stages. This raises also some other questions such as, whether these effects are more prominent at later ages with more prominent social interactions.
- 2) Figures needs more work to be easy to read and understand. Also, better descriptions in the text and figure legends is needed for understanding the figures.
- 3) Selecting 200 random neurons and displaying them is not an analysis but a data visualization tool. I

expected that the authors are expert on such data analysis, and can do much better in analyzing the encoding or spatial features of OXT neurons.

4) The jump from Figure 3 to Figure 4 is a big one. It feels like 2 different stories are merged without sufficient exploration of the second phenomenon (the food uptake). Hence the authors functional work for nociception cannot be generalized to food uptake part of the story. Neither genetic and pharmacological manipulations of the food update can be generalized to the nociception part of the story.

5) The intensity of motor responses induced by nociceptive stimuli, vs any random spontaneous swim bouts are likely very different. Hence, they cannot be interpreted, unless only spontaneous swim bouts with similar intensities to nociceptive induce swim bots are compared, for studying the effect of kin-water in nociception.

6) I found the introduction too short, and the justification of the study is not well aligned with the previous work.

7) Can a group of 3 to 5 individuals can be considered normal condition? Is the effect more prominent, in larger/denser groups?

8) figure 1d) a probability density that goes up to 2?

9) figure 3g) how significant is the difference between the two medians? Of which magnitude? Is it even relevant as an anatomical difference?

Reviewer #2 (Remarks to the Author):

Wee, Song, Nikitchenko and co-authors use larval zebrafish to address how oxytocinergic circuits help shape behavioral outputs. This work is a continuation of published research from the authors that characterized the role of oxytocinergic neurons in regulating responses to noxious stimuli. Here, they extend their analysis to social isolation. Established protocols for assaying neuronal activity with pERK staining revealed that social isolation increases activity in OXT neurons. The presentation of conspecific stimuli (visual or olfactory) was found to suppress isolation induced OXT activity, with olfactory cues providing the strongest reduction. In vivo calcium imaging was conducted to determine how activity in the olfactory bulb and preoptic OXT population is altered by kin, non-kin & adult fish odor cues. Voxels in the OB were identified that were selective for one of the three odor cues or water flow. For each of the odor cues, a subpopulation of OXT neurons responded with enhanced activity while another subpopulation showed suppressed activity. The kin cue produced ~ 4 times the number of suppressed neurons to enhanced neurons, whereas the non-kin cue produced roughly equal numbers. Adult cues produced mostly enhancement. The authors next examined whether the kin odor cue could reduce behavioral outputs associated with a noxious stimulus (optical activation of TRPA1). The presence of the kin cue produced a subtle reduction in the peak tail angle following TRPA1 activation. Regressor analysis of the imaging data found that kin cue suppressed neurons were highly correlated with either the stimulus or stimulus related motor activity. The authors state that the imaging data is consistent with OXT neuron suppression contributing to the observed reduction in TRPA1 mediated nocifensive behavior. A final set of experiments use a paramecia feeding assay to examine the relationship between social modulation of OXT neurons and appetitive behavior. Social isolation was found to reduce feeding and this reduction was rescued by allowing isolated fish to hunt in kin water. Results from experiments where OXT circuits were disrupted by either genetic ablation, pharmacology or mutation show that changes in OXT signaling can explain the feeding differences seen in isolated and grouped animals. The authors conclude that oxytocin circuits in larval zebrafish are modulated by social contexts and that this modulation leads to opposing changes in nocifensive and appetitive behaviours.

Major Comments:

This is a very interesting and well conducted study that extends the understanding of oxytocin signaling in zebrafish to social contexts. It also uncovers some rather surprising findings about larval zebrafish sociality. Zebrafish are not overtly social (shoaling) until ~ 3 weeks of age and it has been

assumed (by this reviewer at least) that the behavior of young larvae is not significantly regulated by social cues other than by avoidance. This work demonstrates that this is not the case. It is also somewhat remarkable that kin water was able to rescue the observed isolation phenotypes. This finding alone will in no doubt spur many future research directions. The work will be of broad interest since it is the first zebrafish research to show that, as in mammals, oxytocin signaling regulates appetitive behaviors. The aforementioned social signal uncovered for young larvae will also be of broad appeal.

- Fig 2 states that the *oxt:Gal4; UAS:GCaMP6s* line weakly labels the PT population. The difference in PT labeling in this line and the *oxt:GFP* line seems apparent when comparing 1C & 2H (PT is mostly missing in 2H). From this comparison, it seems that PT was under sampled in the calcium imaging data in Fig 2 & 3. Is this the case? If so, this should be mentioned.

- For the regressor analysis, how can correlations with the stimulus and stimulus motor response be differentiated given that these events should occur in very tight succession? It would be helpful to have information concerning how reliably optical activation of TRPA1 triggered tail movements.

Minor Comments:

- Figure 1d top right is very difficult to make out. Perhaps there is a better way to distribute the data within 1d?

- It would be helpful to add the name of the mutant to Figure 4g

- Typo Suppl Fig 3 (b) The presence of conspecific cues significantly "decreases" TRPA1-induced velocity changes.

Tod Thiele

Reviewer #3 (Remarks to the Author):

Social deprivation induces nocifensive responses and altered appetite but the underlying mechanism remains unclear (Kikusui, et al., 2006; Higgs & Thomas, 2016). In this very interesting manuscript, Wee and colleagues show that isolated zebrafish larvae have increased oxytocinergic (OXT) neuron activity, which can be inhibited by the presence of social cues from genetically-related animals (kin). Excitingly, they found that kin cues also ameliorate OXT-neuron mediated noxious response and appetite reduction induced by social isolation. The experiments and results described in this manuscript demonstrate critical roles for the oxytocinergic (OXT) circuit in encoding environmental stimulus of social isolation and propose how social stimuli might mediate adaptive behavior.

In general, this is a great piece of work. However, in our opinion, a few pieces of information missing from the manuscript lead to some caveats in the conclusions. Please see comments below for details. We imagine that the authors already have data to address most of our concerns.

Yunlu Zhu, Ph.D. and David Schoppik, Ph.D.

Specific Comments

Line 58. Because the intensity of noxious stimulation modulates OXT neuron activity (Wee, et al., 2019), does the length of isolation also affect neuron activity and/or the population of neurons being activated?

Line 68. "Thus...". Compared to the noxious stimuli which induce more significant activity increase in

OXT neurons (Wee, et al., 2019), social isolation results in a similar, if not greater, activity increase in non-OXT neurons (Suppl. Fig. 1b). It is likely that a large group of non-OXT neurons also encode social isolation/negative valence stimuli. We recommend that the authors soften the claim by taking non-OXT neurons into account.

Fig. 1a, bottom. We find the ROI label misleading. Because the ROIs used for calculation were based on anatomical structures/brain regions and activity of both OXT and non-OXT neurons were measured, the authors should consider using PO and PT for the ROI names instead of OXTpo and OXTpt.

Line 81. Fig1a. Similar to the comment above, because the pERK activity staining is not restricted to OXT neurons, it would be clearer to use PO and PT to describe the outlined region in the legend.

Fig. 1d (top-left) vs Suppl. Fig. 1b (top left). The authors used the same data and statistics for OXT neuron activity in these two panels. However, the probability distribution of [Isolated] group (pink in Fig. 1d) does not align with the [oxt+ (isolated)] group (green in Suppl. Fig. 1b). It seems that the [oxt+ (isolated)] curve in the suppl. fig. is shifted slightly to the right. Please fix.

Line 247-251. How do other conditions, such as non-kin water and adult cues, affect swimming speed and tail bend under Trpa1 stimulation? These results would clarify how environmental stimuli modulate adaptive behavior through the OXT circuit.

Line 327. Because social isolation is known to be associated with increased aversive behavior, do social-isolated larvae have stronger nocifensive response? This can be done by (1) analyzing tethered behavior kinematics of isolated larvae (repeat Fig. 3a to isolated larvae); (2) comparing TRPA1 induced defensive response in isolated vs grouped larvae (repeat Suppl. Fig. 3 to grouped larvae). Because TRPA1 induced aversive behavior is encoded by the OXT circuit (Wee, et al., 2019), results from the proposed experiments may provide direct evidence to determine the role of OXT neurons in encoding negative valence stimuli induced adaptive behavior. If the authors cannot perform such experiments it would be sufficient to address these limitations and propose future work in the discussion.

Line 332. Citation error. None of the conclusions in the citation #22 (Bazhan & Zelena, 2013) supports the claim: "social isolation is associated with feeding behavior in mammals". Perhaps it should be #29?

Line 439. There is no Fig. 3h.

Other minor comments

Scale bars missing for Fig. 1a, Fig. 2, Fig. 3a, Suppl. Fig. 2, Movie 1.

Suppl. Fig. 1b bottom-left panel. Distribution curves of [oxt- (group)] (gray) and [oxt- (non-kin water)] (red) look very similar and almost overlap with each other. Is there a better way to visualize the data?

Suppl. Fig. 1b. line 25. Typo in the figure legend: "In contrast to OXT neurons, non-kin water induced significantly-lower OXT neuron activity relative to non-kin water (**p = 8.7x10⁻²⁵)."¹¹_{SEP}
Figure 2. It seems that the horizontal and vertical perspective of the whole figure is distorted.

Line 526. Typo. Bracket missing for Tg(oxt:GFP)

Reviewer #4 (Remarks to the Author):

The manuscript by Wee et al finds that social isolation modulates 1) behavioral responses to aversive stimuli and 2) feeding behavior. For the first set of behaviors, the authors show correlative evidence for a role of oxytocin neurons. Social isolation activates oxytocin neurons in the zebra fish larvae in a similar fashion as other aversive stimuli. They find that this Oxt activation can be prevented by chemical cues from conspecifics. For the second set of behaviors, the authors show that there is a causal link between the activity of Oxt neurons, oxytocin peptide action, and appetite. Overall, this is a solid study with certain novel findings that would be relevant to the wider scientific community, but I have few concerns that I will list below:

Figure 1:

In Fig. 1a - there seems to be a wide distribution of changes in pERK/tERK ratio following social isolation. How was the 2hr isolation time interval selected? Would the effect be stronger if the isolation was longer?

Fig 1b - the example confocal images are a bit underwhelming, would be great if there were more clear examples

Fig 1c - from the bottom image, there appears to be more significant difference on PT Oxt neurons. Particularly when it comes to the difference between 'isolated+kin water' vs 'isolated+non-kin water'. Perhaps if the comparison was done per region as opposed to all regions together, the data would be stronger. Bottom, it is unclear from legend if the 300 neurons come from each animal or from all animals. If the later is true (as it seems from figure), we need to know how many neurons were selected from each fish (all fish should be equally sampled).

Fig. 1d - The result presentation is not intuitive and it is difficult to follow. Plus, here there seems to be a difference with the 'per fish' results (kin vs non-kin water). This suggests that only in some fish there is a distinction between responses to kin and conspecific water, and perhaps this point should not be emphasized.

Fig. S1 - there seem to be stronger responses in non-Oxt neurons than in Oxt neurons - therefore, why focus on Oxt cells?

Figure 2:

-For calcium imaging, older animals were used than in the previous experiment - this could be confounding

-The olfactory bulb data seems a bit distracting, unless the authors claim that activity patterns in this structure directly dictate/predict the different responses in Oxt neurons.

-It is unfortunate that the Tg(oxt:Gal4; UAS:GCaMP6s) transgenic line does not label Oxt PT neurons, since this population of neurons seemed to have most pERK increases after social isolation

Figure 3:

In this figure, there were no experiments indicating that activity of Oxt neurons is causally linked to behavioral responses to aversive stimuli. Could some of the approaches in Fig. 4 be used in this case too?

Figure 4:

This figure lacks neuronal activity data - we do not know how hunger vs hunger+kin water modulate activity of Oxt neurons. There might be heterogeneity of responses in this case too, and possibly a different spatial distribution than in the case of aversive stim

The manuscript does not identify the chemical cue in the water that might indicate to the larvae the proximity of a kin. Having one or several target chemicals in the conspecific water, or even better in the kin vs non-kin water, would significantly enrich the paper and the influence it has on the thinking in the field.

Response to Reviewers:

Social isolation modulates appetite and defensive behavior via a common oxytocinergic circuit in larval zebrafish

Reviewer #1 (Remarks to the Author):

Here, Wee et al presented a study investigating the role of OXT neurons in regulating animals' responses to appetitive and aversive cues, and how such regulation can be controlled by olfactory social cues. The authors first presented a behavioral assay where, they showed the alterations of brain activity in response to social isolation, using pERK method. The authors argue that a sub population of OXT neurons increase their activity in response to social isolation, and this increase can be dampened by non-kin and kin water from larval fish. Next, using calcium imaging the authors showed that olfactory regions of the zebrafish brain respond to non-kin/kin water from larval fish, and adult fish. Later the authors showed that a sub population of OXT neurons decrease their calcium activity, in response to non-kin/kin water from larval fish, and another population increase their calcium activity when presented with adult fish water. Next, the authors argue that the kin water from larval fish modulate OXT nociceptive responses. Later, the authors studied the effect of social isolation in feeding behaviors, and showed that animals raised in bigger groups eat more. Finally the authors showed that genetic/pharmacological silencing of OXT system alter feeding behavior in isolated animals but not animals raised in groups.

Overall, the behavioral part of this study is interesting and solid.

However, the functional investigations of the phenomenon are rather weak. Especially the results section is not well presented, and most of the functional results are only weakly supported by the presented data and analysis. The figures are confusing and not coherent; hence I find myself reading the text/figures/legends again and again trying to understand how the presented data support authors comments. In general, most functional effects pointed by the neural data are weak and convoluted. Altogether, I find authors hypothesis interesting, but only weakly supported by the presented neural data, which is rather preliminary and not well presented.

We thank the reviewer for their helpful comments and regret the lack of clarity in the figures and their description in the text. In the revision, we have modified the figures (listed after comment #1) and streamlined their presentation. We have also added new evidence for our hypothesis, particularly regarding the circuit link between olfactory and OXT neuronal activity (new Figure 4 of revised manuscript).

Please find below my comments, which I hope will help the authors to improve their study.

1) Figures are very confusing, and authors use several different and often convoluted approaches to present same type of data in different figures. Also, the results are not well described in the text or the figures. I highly recommend the authors to spend some time in preparing better figures and better analysis.

As recommended by the reviewer, we have substantially revised our figures to reduce redundancy, streamlined our analysis methods, and improved on results description in the text and figure legends.

The following main figures were extensively revised:
Figure 1; Figure 2 (old Figure 3); Figure 3 (old Figure 2); Figure 5

Figure 4 now contains new experimental data. In addition, Supplementary Figures 2 - 6, and 8 contain substantial new data and analyses.

2) The link to the olfactory system is weak. How the olfactory system is linked to OXT system?

To address this concern, we have performed additional experiments and analyses that strengthen the link between olfactory input and OXT neuronal activity. We identified a genetically-defined population of GABAergic neurons in the subpallium (Bruckner et al., 2020; Stednitz et al., 2018), that are directly downstream of olfactory bulb output (Biechl et al., 2017; Gerlach and Wullimann, 2021; Miyasaka et al., 2009, 2014; Yaksi et al., 2009) and show that they are responsive to chemical conspecific cues. Using optogenetic stimulation, we further showed that activation of this GABAergic population is sufficient to reduce OXT neuronal activity. These results, presented in the new Figure 4, support our contention that an anatomically- and functionally-defined inhibitory pathway links the olfactory and OXT systems, and also complements other reports in the literature (cited above) of the same subpallial region being involved in social recognition and behavior.

Is it even clear that the presented effect is due to odors?

We agree that we cannot distinguish here ‘odor’ input from other forms of chemosensory input, such as gustatory input. We have, by providing evidence of circuit linkage between olfactory processing and OXT neuron modulation, strengthened the contention that conspecific-conditioned water provides a chemosensory signal for the social regulation of behavior. Thus, we refer to this signal as a “chemosensory cue” throughout the manuscript.

I could imagine that water samples from different groups might have differences in pH, oxygen levels or other factors associated with the presence of animals.

Water-borne chemical agents such as pH or gas (CO₂, O₂) are unlikely to mediate the relevant olfactory input, as the conditioned water was buffered (with pH tested for prior to each experiment) and also equilibrated with room air. Furthermore, the preparation of conditioned water was well-controlled: the number, concentration and time of conditioning was carefully maintained. In calcium imaging experiments, ~300 ul of conditioned water was delivered to the exposed nostrils of the position-fixed animal while a constant perfusion with a larger volume of buffered water (5 ml) was maintained in the chamber.

What content of the fish water carries the social information, and how this information reaches to OXT neurons is a major lacking point.

The chemical nature of the social cue in conditioned water is as yet undetermined, and we have discussed this point in Lines 747-753. In our revision we have focused on providing evidence for the response of the olfactory system to water-borne social cues and its transduction via an inhibitory circuit to OXT neurons, which we show are thus

modulated in ways consistent with behavioral consequences. Specifically, we have added new experimental data showing that the activity of a genetically-defined population of GABAergic neurons in the subpallium, a known target of the olfactory bulb, is modulated by chemosensory input, and that optogenetic stimulation of this population leads to a suppressive effect on OXT neuron activity (Figure 4). Hence, we propose that this subpallial region serves as a circuit link between olfactory input and OXT modulation.

3) The diverse effects on OXT neurons is interesting, but the underlying reasons are not investigated well enough. Why some OXT neurons increase and some decrease their activity in response to odors?

We apologize for not elaborating on this important point. Converging evidence suggests that the OXT population (in larval zebrafish as well as in mammals), is highly heterogeneous in gene expression and projection patterns, and serves multiple roles. As such it is not surprising that the responses of these neurons are similarly heterogeneous in nature, which we tried to describe in detail in our analysis. Nonetheless, we and others (e.g. (Lovett-Barron et al., 2020; Wee et al., 2019)) found that the majority of OXT neurons respond to noxious / aversive stimuli, including social isolation, and that kin water induces a widespread reduction in OXT activity. We now make this point explicitly in the text (Lines 721-731, as well as 803- 819 and in the Discussion Section)

4) The modulation of OXT neurons responses to noxious stimuli by kin-water is very weak as evident by only few selected data points showing such modulation. Better experimental design is necessary to show that such modulation is significant and reproducible.

We hope that our revised Figure 5 makes clearer the extent of modulation of OXT neuron activity by kin water, particularly that the suppressive effect of kin water on TRPA1-responsive OXT neurons is widespread, significant, and reproducible. We do completely agree with the reviewer that, given the heterogeneity of the OXT population, not all nociceptive OXT neurons show such modulation by kin water, and likewise that other neurons in the preoptic area are likely involved in mediating kin water effects on nocifensive behavior, for example, corticotropin-releasing factor-expressing neurons (CRF; see Lovett-Barron et al, 2020). We have now explicitly discussed this in the text (Lines 716-720 and 776-778).

5) I am surprised to see that the authors did not investigate sufficiently the encoding of different water samples (kin, non-kin, adult, etc) in the olfactory or OXT neurons

We apologize for the poor design of the figures. We present now a new and modified set of figure panels that should make much clearer how the different conditioned water samples are encoded by the neurons of the olfactory, subpallial and OXT system.

Minor comments

1) Do the animals have social interactions at this age that is studied? Previous studies from several zebrafish labs suggest that social interactions are rather limited at these developmental stages. This raises also some other questions such as, whether these effects are more prominent at later ages with more prominent social interactions.

The referee is right that social behavior in zebrafish is much more prominent at later ages, and we focused on the larval stage mainly for its optical transparency and amenability towards imaging and circuit manipulation. However, converging evidence does point to an early influence of olfactory, mechanosensory and visual social cues on behavior. This includes a body of work demonstrating early kin recognition and imprinting (Gerlach and Wullimann, 2021; Gerlach et al., 2019), evidence of early visual / mechanosensory-mediated social avoidance which turns into attraction at around 2 weeks (Dreosti et al., 2015; Groneberg et al., 2020; Harpaz et al., 2021), and early social cue mechanosensation (Anneser et al., 2020). Our work contributes to this literature by examining the functional effects of social cue chemosensation on behavior. We have now elaborated on this point in the Introduction (Lines 45-55).

2) Figures needs more work to be easy to read and understand. Also, better descriptions in the text and figure legends is needed for understanding the figures.

We have improved and simplified both the figures and figure legends.

3) Selecting 200 random neurons and displaying them is not an analysis but a data visualization tool. I expected that the authors are expert on such data analysis, and can do much better in analyzing the encoding or spatial features of OXT neurons.

We agree with the reviewer and have added two analyses to aid in data visualization: first, as also suggested by Reviewer 4, we sample the same number of neurons from each fish to prevent over-representation of any individual fish (Figure 1e). Second, we plot a histogram of these active cells along the anterior-posterior axis (Figure 1f), along which there is a clear distribution of OXT neuron cell types.

4) The jump from Figure 3 to Figure 4 is a big one. It feels like 2 different stories are merged without sufficient exploration of the second phenomenon (the food uptake). Hence the authors functional work for nociception cannot be generalized to food uptake part of the story. Neither genetic and pharmacological manipulations of the food update can be generalized to the nociception part of the story.

We agree completely with the reviewer, particularly with the fact that the role of OXT in zebrafish nociception is much better characterized than its effects on feeding. Unlike nociceptive behavior, we are uncertain of the specific pathway via which OXT neurons mediate appetite suppression and hope to follow up on this in future studies. We also agree that the genetic and pharmacological manipulations of feeding cannot be generalized to the nociception part of the story. We had previously demonstrated that OXT neuron activation is sufficient to stimulate, and ablation/loss-of-function sufficient to reduce nociceptive (TRPA1-induced) behaviors, albeit only moderately (Wee et al., 2019). Another subsequent study demonstrated that co-ablation of both OXT and CRF-expressing neurons, but not either alone, was sufficient to abolish responses to heat, salinity, and acidity, confirming that OXT in coordination with another preoptic cell type is crucial for mediating aversive behavior (Lovett-Barron et al., 2020). We agree that this functional work for nociception cannot be generalized to food uptake part of the story, hence we performed functional manipulations in the context of social feeding.

We have highlighted these points in the discussion (lines 771-778).

5) *The intensity of motor responses induced by nociceptive stimuli, vs any random spontaneous swim bouts are likely very different. Hence, they cannot be interpreted, unless only spontaneous swim bouts with similar intensities to nociceptive induce swim bots are compared, for studying the effect of kin-water in nociception.*

The reviewer is right that spontaneous swim bouts may exhibit a different distribution of tail angles than noxious behavior. Hence, we compared the effect of kin water on large vs small-angle bouts and show that specifically, larger-angle responses to TRPA1 stimulation are reduced after kin water exposure, and there was also a significant reduction in the total number of bouts (Figure 5a). Further, even *prior* to TRPA1 stimulation, there was a similar suppressive effect of kin water on the frequency of large angle bouts, although the frequency of such bouts was about 10 fold lower than after TRPA1 stimulation. The reduction in large-angle bout frequency after kin water delivery correlated with the observed reduction in OXT neuron activity prior to TRPA1 stimulation in many OXT neurons (Figure 5b, f), again suggesting that OXT may contribute towards this modulation of behavioral output by kin water.

6) *I found the introduction too short, and the justification of the study is not well aligned with the previous work.*

We agree that the introduction was too brief and incomplete to properly relate the motivation of the experimental work. This has been rectified in the revision (Lines 45-55) in which we summarize the existing literature on zebrafish social behavior.

7) *Can a group of 3 to 5 individuals can be considered normal condition? Is the effect more prominent, in larger/denser groups?*

We agree with the reviewer that a group of 3 or 5 is not a normal condition, and that the observed effects will likely be more prominent in larger groups. In our experiments we wanted to control to the best of our abilities for fish density within a small dish, since higher fish density could be stressful to the fish and also influence behavior (Andersson and Kettunen, 2021; Ribas et al., 2017; Zaki et al., 2021). Hence, we primarily focused on groups with the smallest number / lowest density of fish that demonstrated a significant effect (i.e. group of 3).

8) *figure 1d) a probability density that goes up to 2?*

Although indeed counter-intuitive, the probability density value can increase beyond 1, as it is the area under the curve, rather than the peak of the curve, that should add up to 1. We have now represented Figure 1d using raincloud plots (Allen et al., 2021; Whitaker et al., 2019) which should be clearer to parse than the previous visualization.

9) *figure 3g) how significant is the difference between the two medians? Of which magnitude? Is it even relevant as an anatomical difference?*

We agree with the reviewer that the difference between the medians is small and non-significant. As such we no longer emphasize this point, though the spatial distribution of neurons is still described in Supplementary Figure 8.

We thank the reviewer for their helpful and detailed feedback!

Reviewer #2 (Remarks to the Author):

Wee, Song, Nikitchenko and co-authors use larval zebrafish to address how oxytocinergic circuits help shape behavioral outputs. This work is a continuation of published research from the authors that characterized the role of oxytocinergic neurons in regulating responses to noxious stimuli. Here, they extend their analysis to social isolation. Established protocols for assaying neuronal activity with pERK staining revealed that social isolation increases activity in OXT neurons. The presentation of conspecific stimuli (visual or olfactory) was found to suppress isolation induced OXT activity, with olfactory cues providing the strongest reduction. In vivo calcium imaging was conducted to determine how activity in the olfactory bulb and preoptic OXT population is altered by kin, non-kin & adult fish odor cues. Voxels in the OB were identified that were selective for one of the three odor cues or water flow. For each of the odor cues, a subpopulation of OXT neurons responded with enhanced activity while another subpopulation showed suppressed activity. The kin cue produced ~ 4 times the number of suppressed neurons to enhanced neurons, whereas the non-kin cue produced roughly equal numbers. Adult cues produced mostly enhancement. The authors next examined whether the kin odor cue could reduce behavioral outputs associated with a noxious stimulus (optical activation of TRPA1). The presence of the kin cue produced a subtle reduction in the peak tail angle following TRPA1 activation. Regressor analysis of the imaging data found that kin cue suppressed neurons were highly correlated with either the stimulus or stimulus related motor activity. The authors state that the imaging data is consistent with OXT neuron suppression contributing to the observed reduction in TRPA1 mediated nocifensive behavior. A final set of experiments use a paramecia feeding assay to examine the relationship between social modulation of OXT neurons and appetitive behavior. Social isolation was found to reduce feeding and this reduction was rescued by allowing isolated fish to hunt in kin water. Results from experiments where OXT circuits were disrupted by either genetic ablation, pharmacology or mutation show that changes in OXT signaling can explain the feeding differences seen in isolated and grouped animals. The authors conclude that oxytocin circuits in larval zebrafish are modulated by social contexts and that this modulation leads to opposing changes in nocifensive and appetitive behaviours.

Major Comments:

This is a very interesting and well conducted study that extends the understanding of oxytocin signaling in zebrafish to social contexts. It also uncovers some rather surprising findings about larval zebrafish sociality. Zebrafish are not overtly social (shoaling) until ~ 3 weeks of age and it has been assumed (by this reviewer at least) that the behavior of young larvae is not significantly regulated by social cues other than by avoidance. This work demonstrates that this is not the case. It is also somewhat remarkable that kin water was able to rescue the observed isolation phenotypes. This finding alone will in no doubt spur many future research directions. The work will be of broad interest since it is the first zebrafish research to show that, as in mammals, oxytocin signaling regulates appetitive behaviors. The aforementioned social signal uncovered for young larvae will also be of broad appeal.

We thank the reviewer for their kind comments.

*- Fig 2 states that the *oxt:Gal4; UAS:GCaMP6s* line weakly labels the PT population. The difference in PT labeling in this line and the *oxt:GFP* line seems apparent when comparing 1C & 2H (PT is mostly missing in 2H). From this comparison, it seems that*

PT was under sampled in the calcium imaging data in Fig 2 & 3. Is this the case? If so, this should be mentioned.

Indeed, the PT was under-sampled in our calcium imaging experiments, both because the Gal4 line doesn't label OXT PT very well, and also because many neurons are located more ventrally and posteriorly and hence were difficult to image simultaneously with our current setup. We have highlighted this point in Figure 2a and now explicitly presented PT data where possible (e.g., Figures 2b-c, Figure 4, also Figure 1). We have also elaborated on the possibly differential roles of PT and PO neurons in the discussion (Lines 737-746) and highlighted the under-sampling of PT neurons in our calcium imaging experiments (Line 741).

- For the regressor analysis, how can correlations with the stimulus and stimulus motor response be differentiated given that these events should occur in very tight succession? It would helpful to have information concerning how reliably optical activation of TRPA1 triggered tail movements.

The reviewer is astute in noting the correlation between stimulus and motor regressors. The correlation between stimulus and motor_{stim} regressor ranges from 57-97% depending on the individual fish ($r = 0.84 \pm 0.026$, $n = 16$ fish), and as shown in Figure 5 and Supplementary Figure 8d, there is also tight correlation between each individual neuron's response to both regressors. As such, we have revamped our analysis methods and results section to highlight this point. We hope that our new figures are clearer and better describe the diversity of OXT neuron correlations with both sensory stimuli and motor behavior.

Minor Comments:

- Figure 1d top right is very difficult to make out. Perhaps there is a better way to distribute the data within 1d?

Thank you for the suggestion, we have replotted Figure 1d using a raincloud plot (Allen et al., 2021; Whitaker et al., 2019), which we hope better highlights the differences in pERK intensities.

- It would be helpful to add the name of the mutant to Figure 4g

Thank you, we have implemented this suggestion in new Figure 6g.

- Typo Suppl Fig 3 (b) The presence of conspecific cues significantly "decreases" TRPA1-induced velocity changes.

Thank you for spotting the error, we have made the necessary amendments.

Tod Thiele

Reviewer #3 (Remarks to the Author):

Social deprivation induces nocifensive responses and altered appetite but the underlying mechanism remains unclear (Kikusui, et al., 2006; Higgs & Thomas, 2016). In this very interesting manuscript, Wee and colleagues show that isolated zebrafish larvae have increased oxytocinergic (OXT) neuron activity, which can be inhibited by the presence of social cues from genetically-related animals (kin). Excitingly, they found that kin cues also ameliorate OXT-neuron mediated noxious response and appetite reduction induced by social isolation. The experiments and results described in this manuscript demonstrate critical roles for the oxytocinergic (OXT) circuit in encoding environmental stimulus of social isolation and propose how social stimuli might mediate adaptive behavior.

In general, this is a great piece of work. However, in our opinion, a few pieces of information missing from the manuscript lead to some caveats in the conclusions. Please see comments below for details. We imagine that the authors already have data to address most of our concerns.

Yunlu Zhu, Ph.D. and David Schoppik, Ph.D.

We thank the reviewers for their helpful comments.

Specific Comments

Line 58. Because the intensity of noxious stimulation modulates OXT neuron activity (Wee, et al., 2019), does the length of isolation also affect neuron activity and/or the population of neurons being activated?

In some pilot experiments we conducted, we noted that the pERK experiments suggested that the enhancement of OXT neuron activity during social isolation was reliable at the 2-hour time point, and did not further increase at 4 hours, hence we utilized this time point for most experiments (See **Response to Reviewers Figure 1** at end of this document).

Further, in our calcium imaging data, we did not observe any consistent changes in OXT neuron suppression by kin water with isolation times between 30 min to 4 hours (now plotted in Supplementary Figure 2a, and highlighted in lines 191-192)

Line 68. “Thus...”. Compared to the noxious stimuli which induce more significant activity increase in OXT neurons (Wee, et al., 2019), social isolation results in a similar, if not greater, activity increase in non-OXT neurons (Suppl. Fig. 1b). It is likely that a large group of non-OXT neurons also encode social isolation/negative valence stimuli. We recommend that the authors soften the claim by taking non-OXT neurons into account.

We completely agree with the reviewers' recommendation. We have now addressed this point in lines 102-104 of the results section and lines 716-720 as well as 775-778 in the discussion.

Fig. 1a, bottom. We find the ROI label misleading. Because the ROIs used for calculation were based on anatomical structures/brain regions and activity of both OXT

and non-OXT neurons were measured, the authors should consider using PO and PT for the ROI names instead of OXT_{po} and OXT_{pt}.

Thank you, we have implemented the suggested changes.

Line 81. Fig1a. Similar to the comment above, because the pERK activity staining is not restricted to OXT neurons, it would be clearer to use PO and PT to describe the outlined region in the legend.

Thank you, we have implemented the suggested changes.

Fig. 1d (top-left) vs Suppl. Fig. 1b (top left). The authors used the same data and statistics for OXT neuron activity in these two panels. However, the probability distribution of [Isolated] group (pink in Fig. 1d) does not align with the [oxt+ (isolated)] group (green in Suppl. Fig. 1b). It seems that the [oxt+ (isolated)] curve in the suppl. fig. is shifted slightly to the right. Please fix.

Thank you for the astute observation. We have replotted Figure 1d and Supplementary Figure 1b using the raincloud plot function to fix this issue.

Line 247-251. How do other conditions, such as non-kin water and adult cues, affect swimming speed and tail bend under Trpa1 stimulation? These results would clarify how environmental stimuli modulate adaptive behavior through the OXT circuit.

This is indeed an interesting question; we unfortunately do not have experiments regarding the effects of non-kin or adult cues on TRPA1-induced behavioral responses. We hope to continue to explore the distinctions between these cues in future nocifensive (or feeding) behavioral experiments. In the meantime, we have highlighted this shortcoming in the discussion (lines 732 to 735).

Line 327. Because social isolation is known to be associated with increased aversive behavior, do social-isolated larvae have stronger nocifensive response? This can be done by (1) analyzing tethered behavior kinematics of isolated larvae (repeat Fig. 3a to isolated larvae); (2) comparing TRPA1 induced defensive response in isolated vs grouped larvae (repeat Suppl. Fig. 3 to grouped larvae). Because TRPA1 induced aversive behavior is encoded by the OXT circuit (Wee, et al., 2019), results from the proposed experiments may provide direct evidence to determine the role of OXT neurons in encoding negative valence stimuli induced adaptive behavior. If the authors cannot perform such experiments it would be sufficient to address these limitations and propose future work in the discussion.

We agree that these are good suggestions, however due to practical limitations we have not been able to perform these experiments. For (1), all the fish for Figure 5's experiments were isolated for at least 2 hours. In the future we could examine the effects of varying lengths of isolation duration on nocifensive behavior.

For (2), we did consider performing these experiments in the past, but were concerned about the potential confound of testing fish in a group, as their behavioral responses may also influence each other's (e.g. fish colliding into one another or indirectly triggering escape behaviors due to lateral line stimulation). Hence, we resorted to using kin water as a "cleaner" experiment.

As suggested by the reviewers, we have addressed these limitations in the discussion (lines 767-771).

Line 332. Citation error. None of the conclusions in the citation #22 (Bazhan & Zelena, 2013) supports the claim: “social isolation is associated with feeding behavior in mammals”. Perhaps it should be #29?

The reviewer is right that the correct reference is Higgs and Thomas (2016). Thank you for spotting this error!

Line 439. There is no Fig. 3h.

The reference was meant for Figure 4h, and we have now checked through our figure references in the manuscript.

Other minor comments

Scale bars missing for Fig. 1a, Fig. 2, Fig. 3a, Suppl. Fig. 2, Movie 1.

Thank you, we have added scale bars to all images.

Suppl. Fig. 1b bottom-left panel. Distribution curves of [oxt- (group)] (gray) and [oxt- (non-kin water)] (red) look very similar and almost overlap with each other. Is there a better way to visualize the data?

We have re-plotted the results using raincloud plots.

*Suppl. Fig. 1b. line 25. Typo in the figure legend: “In contrast to OXT neurons, non-kin water induced significantly-lower OXT neuron activity relative to non-kin water (**p = 8.7×10^{-25}).”*

We have re-written the legends for Supplementary Figure 1.

Figure 2. It seems that the horizontal and vertical perspective of the whole figure is distorted.

Thank you, this was caused by embedding and reshaping the figures in the text and the original figures should not be distorted.

Line 526. Typo. Bracket missing for Tg(oxt:GFP)

Thank you, we have corrected the error.

Reviewer #4 (Remarks to the Author):

The manuscript by Wee et al finds that social isolation modulates 1) behavioral responses to aversive stimuli and 2) feeding behavior. For the first set of behaviors, the authors show correlative evidence for a role of oxytocin neurons. Social isolation activates oxytocin neurons in the zebra fish larvae in a similar fashion as other aversive stimuli. They find that this Oxt activation can be prevented by chemical cues from conspecifics. For the second set of behaviors, the authors show that there is a causal link between the activity of Oxt neurons, oxytocin peptide action, and appetite. Overall, this is a solid study with certain novel findings that would be relevant to the wider scientific community, but I have few concerns that I will list below:

We thank the reviewer for their positive comments.

Figure 1:

In Fig. 1a - there seems to be a wide distribution of changes in pERK/tERK ratio following social isolation. How was the 2hr isolation time interval selected? Would the effect be stronger if the isolation was longer?

As indicated also in our response to Reviewer 3's first comment, our preliminary pERK experiments suggested that the difference in OXT neuron activity during social isolation started to be prominent at the 2 hour time point, and there no obvious increase after 2 hours, hence we selected this for future experiment (see **Response to Reviewers Figure 1** at end of this document).

Further, in our calcium imaging data, we did not observe any consistent changes in OXT neuron suppression by kin water with isolation times between 30 min to 4 hours (now plotted in Supplementary Figure 2a, and highlighted in lines 191-192)

Fig 1b – the example confocal images are a bit underwhelming, would be great if there were more clear examples.

We have replaced Figure 1b with a clearer image and improved on data visualization of our pERK analysis in Figure 1. However, we also note that pERK is not as sensitive as calcium imaging as a quantitative (or qualitative) tool, since only the neurons of highest activity would be distinguishable from background staining. Further it is a less direct readout of neuronal activity than calcium fluorescence. Hence, we have complemented our pERK with calcium imaging experiments to provide better temporal resolution and support for our arguments.

Fig 1c – from the bottom image, there appears to be more significant difference on PT Oxt neurons. Particularly when it comes to the difference between 'isolated+kin water' vs 'isolated+non-kin water'. Perhaps if the comparison was done per region as opposed to all regions together, the data would be stronger. Bottom, it is unclear from legend if the 300 neurons come from each animal or from all animals. If the later is true (as it seems from figure), we need to know how many neurons were selected from each fish (all fish should be equally sampled).

Thank you for this great suggestion -- we have now performed our analysis per region (PO or PT) and also with equal sampling of neurons across all fish, and identified some differences between their responses to conspecific cues. As indicated also in our

response to Reviewer 2's first comment, we now explicitly separate PT responses where possible (e.g., Figures 2b-c, Figure 4) have also elaborated on the possibly differential roles of PT and PO neurons in the discussion (Lines 737-746).

Fig. 1d – The result presentation is not intuitive and it is difficult to follow. Plus, here there seems to be a difference with the 'per fish' results (kin vs non-kin water). This suggests that only in some fish there is a distinction between responses to kin and conspecific water, and perhaps this point should not be emphasized.

We apologize for the confusion and hope we have improved on presentation in the revised manuscript. The reviewer is right that, given the variability across fish, the differences between kin and non-kin water effects on OXT activity may be difficult to discern via pERK staining, especially when a single value (mean pERK intensity) is used to summarize the activities of the entire OXT population. Calcium imaging of the OXT population, which provides higher resolution, supports our pERK results, however again, the distinction between kin and non-kin cues is smaller compared to the distinction between conspecific larval and adult cues, or between conspecific larval cues and water flow.

Further, we have also not compared kin and non-kin effects on behavioral output, and hence have not demonstrated the impact of such differences in modulation behaviorally.

Overall, we agree that the main message of our manuscript should focus on the effects of conspecific social cues, rather than between kin and non-kin water responses, and we have rewritten the manuscript to highlight this fact (e.g., Lines 732 to 737 in the Discussion)

However, we have still reported the distinctions we see between between kin and non-kin responses across the olfactory bulb (Figure 3), subpallial (Figure 4) and OXT populations (Figure 1-2), as they are significant, reproducible across experiments, and complement a wider body of literature already demonstrating the ability of young zebrafish to discriminate these signals (Gerlach and Wullimann, 2021). Notably, kin odors were much more strongly represented in telencephalic nuclei (Figure 4a Supplementary Figures 3-5), with fewer neurons responding to non-kin relative to kin or adult water. Further, we have observed some differences of OXT_{PO} and OXT_{PT}, as well as non-OXT neuron responses to kin and non-kin cues (Figure 1, Supplementary Figure 1), which could inform future follow-up studies.

Fig. S1 – there seem to be stronger responses in non-Oxt neurons than in Oxt neurons – therefore, why focus on Oxt cells?

The reviewer is right that many non-OXT neurons in the preoptic area have strong effects on isolation. The focus on OXT was for two reasons, first that we and others have already extensively characterized its anatomy and function, especially its role in aversive nocifensive behavior and second, because of its known relevance to social behavior in mammals.

Interestingly, a study published in 2020 (Lovett-Barron et al., 2020) characterized the role of multiple neuropeptidergic populations in the preoptic area and demonstrated that many are responsive to aversive stimuli and correlate with locomotor activity. In particular, optogenetic activation of either OXT and CRF-expressing neurons, but not

somatostatin-expressing neurons, was sufficient to drive locomotion, and only co-ablation of these two populations could abolish stressor-induced locomotion (they did not look at noxious stimuli). Hence, CRF-expressing neurons could be a subject of our future focus. We have now reiterated the likely role of non-OXT neurons in the text (lines 716-720, 776-778 in the Discussion).

Figure 2:

-For calcium imaging, older animals were used than in the previous experiment – this could be confounding

We agree with the reviewer that the age range of calcium imaging versus pERK experiments do not completely match. However, when segregating fish by age in our calcium imaging experiments, we did not observe any obvious differences in the effect of kin water on OXT activity. These results are now plotted in Supplementary Figure 2b and highlighted in the text (Lines 192-193).

-The olfactory bulb data seems a bit distracting, unless the authors claim that activity patterns in this structure directly dictate/predict the different responses in Oxt neurons.

Thank you for this comment. In the past few months, we have worked on better delineating the relationship between olfactory bulb responses and OXT neuron responses, in particular by identifying a GABAergic subpallial circuit that is activated by conspecific cues and sufficient to suppress OXT neuron and also neuropil activity (Figure 4, Supplementary Figures 4-6). We hope these experiments provide better context for the olfactory bulb imaging data (Figure 3, Supplementary Figure 3).

*-It is unfortunate that the *Tg(oxt:Gal4; UAS:GCaMP6s)* transgenic line does not label Oxt PT neurons, since this population of neurons seemed to have most pERK increases after social isolation.*

Indeed, as also noted in our response to Reviewer 2's first comment, the PT was under-sampled in our study, both because the Gal4 line doesn't label the PT very well and because many neurons are located more ventrally and posteriorly and were difficult to image simultaneously with our current setup. We have now presented PT data where possible (e.g., Figure 2b-c, Figure 4) and have also elaborated on the possibly differential roles of PT and PO neurons in the discussion (Lines 737-746).

Figure 3:

In this figure, there were no experiments indicating that activity of Oxt neurons is causally linked to behavioral responses to aversive stimuli. Could some of the approaches in Fig. 4 be used in this case too?

We agree that we have not explicitly demonstrated the functional necessity of OXT neurons in mediating the social buffering effects of kin cues on nociceptive behaviors. In our previous manuscript (Wee et al, 2019) characterizing OXT neuron role in nociception, we demonstrated that OXT neuron ablation and mutagenesis could reduce nociceptive responses, however the effect was moderate, and we did not explore the role of social cues. A subsequent manuscript by Lovett-Barron (2020) further demonstrated that only co-ablating OXT along with the CRF-expressing in the PO could abolish aversive responses, again supporting the idea that multiple preoptic

neuropeptidergic populations may have overlapping / redundant functions in mediating aversive behavior.

We have now explicitly highlighted this point in the discussion (Lines 771-778).

Figure 4:

This figure lacks neuronal activity data – we do not know how hunger vs hunger+kin water modulate activity of Oxt neurons. There might be heterogeneity of responses in this case too, and possibly a different spatial distribution than in the case of aversive stim

We agree completely with the reviewer. The pERK experiments in Figure 1 were performed in a fed state and the interaction between starvation and isolation state may be heterogenous and more complicated. However, in our preliminary experiments (see **Response to Reviewers Figure 1**) we did examine the effects of isolation in different feeding states, and consistently observed an effect of isolation and conspecific cues on OXT neuron activity in both hunger and satiety. Further experiments comparing OXT activity and the effects of isolation / kin water across these states would be an interesting subject for follow-up on.

The manuscript does not identify the chemical cue in the water that might indicate to the larvae the proximity of a kin. Having one or several target chemicals in the conspecific water, or even better in the kin vs non-kin water, would significantly enrich the paper and the influence it has on the thinking in the field.

Unfortunately, we did not attempt to identify the cues in the water that comprise the kin / conspecific cues, though it is indeed a promising avenue for follow-up research. There have been previous attempts to characterize the chemical nature of kin cues. One study has shown that larval zebrafish imprint onto siblings that express the same MHC class II alleles, suggesting that MHC peptide ligands may be one of the sources of kin recognition (Hinz et al., 2013). Bile acids present in fish excrement have also been proposed to be a source of social recognition (Cong et al., 2019; Kermen et al., 2020). Future studies will be necessary to pinpoint the exact sources of kin information and how they may be encoded by the fish olfactory system. We have discussed some of these possibilities in lines 747-753 of the Discussion to motivate future work in the field.

Response to Reviewers Figure 1: (a) and (b) are time course experiments we previously conducted, where we compared the effects of 30 min, 2 hr, and 4 hr starvation on OXT neuron pERK intensities of isolated (pink) or fish maintained in groups of 3. These were carried out either in the presence (a) or absence (b) of food. At the 30 min time point, OXT activity trended lower in the isolated condition, and there was no further enhancement of OXT activity after 4 hours of isolation. (c) In another preliminary experiment, OXT pERK intensities were measured in fish that had been isolated, maintained in groups of 3, or isolated and exposed to kin or non-kin water. The fish were fixed after 2 hours of starvation and 30 minutes of feeding, to simulate the conditions in which the food intake experiments were carried out. We observed the same trends with the experiments done in fed fish shown in Figure 1 of our manuscript.

References:

- Allen, M., Poggiali, D., Whitaker, K., Marshall, T.R., van Langen, J., and Kievit, R.A. (2021). Raincloud plots: a multi-platform tool for robust data visualization. *Wellcome Open Res.* 4, 63.
- Andersson, M., and Kettunen, P. (2021). Effects of Holding Density on the Welfare of Zebrafish: A Systematic Review. *Zebrafish*.
- Anneser, L., Alcantara, I.C., Gemmer, A., Mirkes, K., Ryu, S., and Schuman, E.M. (2020). The neuropeptide Pth2 dynamically senses others via mechanosensation. *Nature* 588, 653–657.
- Biechl, D., Tietje, K., Ryu, S., Grothe, B., Gerlach, G., and Wullimann, M.F. (2017). Identification of accessory olfactory system and medial amygdala in the zebrafish. *Sci. Rep.* 7, 44295.
- Bruckner, J.J., Stednitz, S.J., Grice, M.Z., Larsch, J., Tallafuss, A., Washbourne, P., and Eisen, J.S. (2020). The microbiota promotes social behavior by neuro-immune modulation of neurite complexity.
- Cong, X., Zheng, Q., Ren, W., Chéron, J.-B., Fiorucci, S., Wen, T., Zhang, C., Yu, H., Golebiowski, J., and Yu, Y. (2019). Zebrafish olfactory receptors ORAs differentially detect bile acids and bile salts. *J. Biol. Chem.* 294, 6762–6771.
- Dreosti, E., Lopes, G., Kampff, A.R., and Wilson, S.W. (2015). Development of social behavior in young zebrafish. *Front. Neural Circuits* 9, 39.
- Gerlach, G., and Wullimann, M.F. (2021). Neural pathways of olfactory kin imprinting and kin recognition in zebrafish. *Cell Tissue Res.* 383, 273–287.
- Gerlach, G., Tietje, K., Biechl, D., Namekawa, I., Schalm, G., and Sulmann, A. (2019). Behavioural and neuronal basis of olfactory imprinting and kin recognition in larval fish. *J. Exp. Biol.* 222.
- Groneberg, A.H., Marques, J.C., Martins, A.L., Diez Del Corral, R., de Polavieja, G.G., and Orger, M.B. (2020). Early-Life Social Experience Shapes Social Avoidance Reactions in Larval Zebrafish. *Curr. Biol.* 30, 4009–4021.e4.
- Harpaz, R., Nguyen, M.N., Bahl, A., and Engert, F. (2021). Precise visuomotor transformations underlying collective behavior in larval zebrafish.
- Hinz, C., Namekawa, I., Namekawa, R., Behrmann-Godel, J., Oppelt, C., Jaeschke, A., Müller, A., Friedrich, R.W., and Gerlach, G. (2013). Olfactory imprinting is triggered by MHC peptide ligands. *Sci. Rep.* 3, 2800.
- Kermen, F., Darnet, L., Wiest, C., Palumbo, F., Bechert, J., Uslu, O., and Yaksi, E. (2020). Stimulus-specific behavioral responses of zebrafish to a large range of odors exhibit individual variability. *BMC Biol.* 18, 66.
- Lovett-Barron, M., Chen, R., Bradbury, S., Andalman, A.S., Wagle, M., Guo, S., and Deisseroth, K. (2020). Multiple convergent hypothalamus–brainstem circuits drive defensive behavior. *Nat. Neurosci.*

Miyasaka, N., Morimoto, K., Tsubokawa, T., Higashijima, S.-I., Okamoto, H., and Yoshihara, Y. (2009). From the olfactory bulb to higher brain centers: genetic visualization of secondary olfactory pathways in zebrafish. *J. Neurosci.* 29, 4756–4767.

Miyasaka, N., Arganda-Carreras, I., Wakisaka, N., Masuda, M., Sümbül, U., Seung, H.S., and Yoshihara, Y. (2014). Olfactory projectome in the zebrafish forebrain revealed by genetic single-neuron labelling. *Nat. Commun.* 5, 3639.

Ribas, L., Valdivieso, A., Díaz, N., and Piferrer, F. (2017). Appropriate rearing density in domesticated zebrafish to avoid masculinization: links with the stress response. *J. Exp. Biol.* 220, 1056–1064.

Stednitz, S.J., McDermott, E.M., Ncube, D., Tallafuss, A., Eisen, J.S., and Washbourne, P. (2018). Forebrain Control of Behaviorally Driven Social Orienting in Zebrafish. *Curr. Biol.* 28, 2445–2451.e3.

Wee, C.L., Nikitchenko, M., and Wang, W.C. (2019). Zebrafish oxytocin neurons drive nocifensive behavior via brainstem premotor targets. *Nature Neuroscience.*

Whitaker, K., TomRhysMarshall, van Mourik, T., Martinez, P.A., Poggiali, D., Ye, H., and MariusKlug (2019). RainCloudPlots/RainCloudPlots: WellcomeOpenResearch.

Yaksi, E., von Saint Paul, F., Niessing, J., Bundschuh, S.T., and Friedrich, R.W. (2009). Transformation of odor representations in target areas of the olfactory bulb. *Nat. Neurosci.* 12, 474–482.

Zaki, H., Lushi, E., and Severi, K.E. (2021). Larval zebrafish exhibit collective motion behaviors in constrained spaces.

REVIEWER COMMENTS

Reviewer #1 (Remarks to the Author):

I thank the authors for this revised manuscript that is substantially improved. Authors did a series of new experimentation, which formed a better link between the olfactory components of OXT neuron modulation, and underlying circuit path. I especially liked the imaging experiments of ventral forebrain and follow up optogenetic circuit mapping that link odor related signals to OXT neuron activity. These are challenging experiments but certainly added a lot of strength to the main message.

I support the publication of this manuscript in Nature Communications.

My only small advice is that the figures are rather heavy, and hence, not easy to grasp. If there is any way to organize/represent these data sets in a more compact way, it will help the readers to grasp the message easier.

On an additional note, I still think that the feeding part of the story (Figure 6) is a bit out place and transition is abrupt. Moreover, the link of the presented data to appetite weaker than defensive behaviors. This is up to the authors and the editors if it is necessary to add that part (Figure 6) to this manuscript. I believe this is a fantastic study even without the feeding part. Leaving feeding part out will results in a more compact and stronger manuscript.

Reviewer #2 (Remarks to the Author):

The authors substantially revised the manuscript and have addressed all of the issues I raised. I find the new link between the subpallial GABAergic population and OXT neurons to be a strong addition. The new text, figures and experiments make the work much clearer and add to its overall significance.

Reviewer #3 (Remarks to the Author):

In this study, Wee et al. demonstrate critical roles of the zebrafish OXT circuit in encoding environmental stimulus of social isolation. In the revised manuscript, the authors have taken a step forward and identified a population of GABAergic subpallial neurons linking the olfactory signal to OXT neurons. In general, the manuscript is clearly written and the results of the experiments justify the conclusions. I only have a few minor comments regarding the optogenetic experiment. Please see below for details.

Yunlu Zhu, Ph.D

I found the optogenetic activation paradigm a little confusing. Figure 4g shows averaged traces of 17-30 cells with multiple laser stimulations in each trial while Figure 4h demonstrates averaged responses of hundreds of neurons. Was the data in Figure 4h and Supplementary Figure 6D collected using the same paradigm as shown in Figure 4g? If so, how many stimulations were in each trial and how many repeats were performed? How are the example traces in Figure 4g selected? Does each row of the raster plots in Supplemental Figure 6d represent averaged activity of individual neurons? If so, how many repeats were there for each neuron? I think these experimental details are necessary and will help with better understanding of the results.

Figure 4g. Line 369. The authors marked OXT neuropil activity that was associated with laser stimulation (arrows) in the bottom panels. However, to my eyes, there are more events like this in both the negative control (black) and the experimental example (red). I suggest the authors label all such activities to avoid ambiguity.

Figure 4h. To me, It is undoubtedly true that optogenetic activation of aSPa neurons suppresses OXT-PO neuron activity. However, I think these results would further benefit from clarity in how the neural/neuropil activities are compared between the control and activation responses. For aSPa and OXT cell body activities, there are dramatic reductions in the peak amplitudes so I assume the peak dF/F were used for quantification. However, for the OXT neuropil, it is not clear how the authors quantified the responses. Please clarify.

Supplementary Figure 6D demonstrates response of individual neurons. I found it difficult to compare between the stimulation groups and the controls due to their drastically different sample size (300 vs 90 for PO cells, 120 vs 10 for PT cells). The authors may try stretching the negative data so that the raster plots can be in the same size and labeling the y axis in "percentage".

Figure 1e. The authors have improved their visualization of OXT neuron distribution. My only other suggestion is to replace the annotation ("highest/lowest pERK signal") with a color spectrum showing corresponding normalized pERK values (similar to that in Figure 2a). Same for Supplementary Figure 6b, bottom.

Reviewer #4 (Remarks to the Author):

For this revised manuscript, the authors performed additional experiments to show a potential circuit mechanism for how conspecific cues in the water might control activity of oxytocin neurons. This is a welcome addition to the paper. However, I would have liked to see a stronger mechanistic link. First of all, the decision to investigate γ 321 neurons is surprising, as the authors cite previous work that shows these neurons are 'implicated in visually-mediated social behavior' (line 340). In their first figure, they show that oxytocin neurons do not respond to visual social cues. Therefore, the choice to focus on this population seems surprising. Second, I would have liked to see an experiment like the one in Figure 2a,b but with suppression of γ 321 neurons – that would indicate if the suppression of OT cells by kin cue requires γ 321 activity. Nevertheless, the current version does give a more mechanistic understanding of social buffering in fish. Given this, I really think the feeding data in Fig. 6 belong to a separate manuscript.

Moreover, I have some additional comments related to figures:

Analysis in Fig. 1c and d :

Why isn't the pERK/tERK used here? It would make more sense I think to use that ratio for normalizing to the control group, given how variable staining can be from specimen to specimen. Also, the statistical analysis does not seem to be corrected – please correct for multiple comparison, for example FDR correction.

Fig. 2b,c,d – I only see descriptive stats for these plots, there is no significance analysis as far as I can tell. I believe at least one of the reviewers required proper stats for this figure

Fig. 2e is not a clear representation of data. If anything, it's confusing. Why not use pie

charts or Venn diagrams to more clearly show the overlap?

Fig. 4c: how do we know the oxytocin fibers are dendrites and not axons?

Fig. 3e: it seems that clusters 3,4,7, have increased responses to both 'kin' and 'adult'. However, the oxytocin cells have distinct responses to these cues. How would the circuit proposed work to produce these distinct responses?

Fig. 3h: please specify the statistical test used.

What do the authors mean by this phrase: 'Across a range of thresholds, the fraction of OXT-positive neurons whose activities were suppressed by larval kin- conditioned water was consistently higher than the fraction of neurons that were activated'? what is that range of thresholds exactly?

What other epochs do the authors refer to here: 'Similarly, in 8-10 dpf tethered fish, kin water significantly reduced the frequency of large-angle swims in tethered fish after kin cue delivery and TRPA1 stimulation but not during other epochs (Figure 5a, Supplementary Figure 8a).'? I hope they don't refer to 'pre-cue' as that is not a logical statement, right? My understanding is that the cue represents the kin water, so I would not expect to see an effect of kin water before this is given.... Am I missing something here?

Fig. 5f: why was a one-sided test used here? Shouldn't this be two-sided?

For all figures, please make the font size within panels bigger.

REVIEWER COMMENTS & RESPONSE

Reviewer #1 (Remarks to the Author):

I thank the authors for this revised manuscript that is substantially improved. Authors did a series of new experimentation, which formed a better link between the olfactory components of OXT neuron modulation, and underlying circuit path. I especially liked the imaging experiments of ventral forebrain and follow up optogenetic circuit mapping that link odor related signals to OXT neuron activity. These are challenging experiments but certainly added a lot of strength to the main message. I support the publication of this manuscript in Nature Communications.

We are pleased that the reviewer is satisfied with the revisions.

My only small advice is that the figures are rather heavy, and hence, not easy to grasp. If there is any way to organize/represent these data sets in a more compact way, it will help the readers to grasp the message easier.

We appreciate the reviewer's feedback and have attempted to the best of our abilities to simplify some of the figures, particularly Figures 1 and 2. The subsequent figures, while heavy, convey essential information, and so we have not been able to simplify them further, though we have enhanced presentation through some resizing of fonts and panels.

On an additional note, I still think that the feeding part of the story (Figure 6) is a bit out place and transition is abrupt. Moreover, the link of the presented data to appetite weaker than defensive behaviors. This is up to the authors and the editors if it is necessary to add that part (Figure 6) to this manuscript. I believe this is a fantastic study even without the feeding part. Leaving feeding part out will results in a more compact and stronger manuscript.

We appreciate the reviewer's point. It is true that we have yet to connect the OXT circuitry to targets regulating feeding, though they do send extensive projections into the hypothalamus (Herget et al., 2017) On the other hand, including appetite regulation along with nociceptive responses supports the proposition that social isolation regulates a set of behaviors, which provides a context in which to understand the diversity of OXT neuron odor responses. The modulation of multiple behaviors justifies comparison to the hub and spoke model that has been identified in other organisms (Macosko et al., 2009). Further, it ties our findings in zebrafish to the important appetite-regulating role of OXT in mammals. We thus feel that its inclusion adds a valuable dimension to the manuscript.

Reviewer #2 (Remarks to the Author):

The authors substantially revised the manuscript and have addressed all of the issues I raised. I find the new link between the subpallial GABAergic population and OXT neurons to be a strong addition. The new text, figures and experiments make the work much clearer and add to its overall significance.

We thank the reviewer for the kind comments.

Reviewer #3 (Remarks to the Author):

In this study, Wee et al. demonstrate critical roles of the zebrafish OXT circuit in encoding environmental stimulus of social isolation. In the revised manuscript, the authors have taken a step forward and identified a population of GABAergic subpallial neurons linking the olfactory signal to OXT neurons. In general, the manuscript is clearly written and the results of the experiments justify the conclusions. I only have a few minor comments regarding the optogenetic experiment. Please see below for details.

Yunlu Zhu, Ph.D

Thank you for your helpful feedback. We hope this revised version has addressed all your concerns.

I found the optogenetic activation paradigm a little confusing. Figure 4g shows averaged traces of 17-30 cells with multiple laser stimulations in each trial while Figure 4h demonstrates averaged responses of hundreds of neurons. Was the data in Figure 4h and Supplementary Figure 6D collected using the same paradigm as shown in Figure 4g? If so, how many stimulations were in each trial and how many repeats were performed?

Indeed, the same or very similar paradigms were used in 4g and 4h, and supplementary Figure 6d. Briefly, we used 7 to 14 stimulations per fish, 60 s -120 s inter stimulus interval. We have now added a more detailed description in the Methods section (also see below).

How are the example traces in Figure 4g selected?

The traces were chosen because they were typical of what we usually observed in such experiments. Since we used a range of inter-stimulus intervals in our experiments, we eventually selected traces that had the same inter-stimulus intervals for easier visualization, and also a fish in which imaging was performed both in the y321 forebrain region and the OXT region.

Does each row of the raster plots in Supplemental Figure 6d represent averaged activity of individual neurons? If so, how many repeats were there for each neuron? I think these experimental details are necessary and will help with better understanding of the results.

Yes, the activity of neurons across all stimulations was averaged to produce the raster plot. We have now added a table to show the number of repeats per neuron and interstimulus intervals in the Methods Section, reproduced here.

Bold = same fish was tested in under both protocols

	y321 ReaChR (record y321)	y321 no ReaChR (record y321)	y321, oxt ReaChR (record OXT)	Negative Control
1	7 stims / 180 s	10 stims, 60 s	11 stims / 180 s	10 stims / 180 s
2	6 stims / 60 s	14 stims, 60 s	10 stims / 180 s	7 stims / 120 s
3	6 stims / 120 s	12 stims, 60 s	13 stims / 180 s	12 stims / 60 s
4	7 stims / 120 s	14 stims, 60 s	13 stims / 120 s	13 stims / 60 s
5	7 stims / 120 s		13 stims / 120 s	12 stims / 60 s
6	7 stims / 120 s		13 stims / 120 s	
7	9 stims / 60 s		14 stims / 60 s	
8	8 stims / 60 s		14 stims / 60 s	
9	8 stims / 60 s		13 stims / 60 s	
10	7 stims / 60 s		14 stims / 60 s	
11	7 stims / 60 s		13 stims / 60 s	
12	7 stims / 60 s		14 stims / 60 s	
13	6 stims / 60 s		12 stims / 60 s	
14	6 stims / 60 s		8 stims / 60 s	
15	10 stims / 60 s		7 stims / 60 s	
16			12 stims / 60 s	
17			10 stims / 60 s	
18			10 stims / 60 s	

Figure 4g. Line 369. The authors marked OXT neuropil activity that was associated with laser stimulation (arrows) in the bottom panels. However, to my eyes, there are more events like this in both the negative control (black) and the experimental example (red). I suggest the authors label all such activities to avoid ambiguity.

Thank you, we have now labeled all events that occur in sync with light stimulus.

Figure 4h. To me, It is undoubtedly true that optogenetic activation of aSPa neurons suppresses OXT-PO neuron activity. However, I think these results would further benefit from clarity in how the neural/neuropil activities are compared between the control and activation responses. For aSPa and OXT cell body activities, there are dramatic reductions in the peak amplitudes so I assume the peak dF/F were used for quantification. However, for the OXT neuropil, it is not clear how the authors quantified the responses. Please clarify.

The statistics were calculated by comparing the integrated (sum) activity across the post-stimulus period highlighted in Figure 4h. However we note the reviewers' point that for OXT neuropil, there are two separate epochs involving both a "dip" followed by "rebound

activity”, hence we have now calculated them separately and also indicated on the figure the period over which df/f signals were compared, and find that y321 activation significantly modulates OXT neuropil activity in both epochs.

Supplementary Figure 6D demonstrates response of individual neurons. I found it difficult to compare between the stimulation groups and the controls due to their drastically different sample size (300 vs 90 for PO cells, 120 vs 10 for PT cells). The authors may try stretching the negative data so that the raster plots can be in the same size and labeling the y axis in “percentage”.

Thank you for the great suggestion. We have decided to represent the plots both ways for better visualization.

Figure 1e. The authors have improved their visualization of OXT neuron distribution. My only other suggestion is to replace the annotation (“highest/lowest pERK signal”) with a color spectrum showing corresponding normalized pERK values (similar to that in Figure 2a). Same for Supplementary Figure 6b, bottom.

Thank you for the comment, we have added a colorbar corresponding to normalized pERK intensity.

Reviewer #4 (Remarks to the Author):

For this revised manuscript, the authors performed additional experiments to show a potential circuit mechanism for how conspecific cues in the water might control activity of oxytocin neurons. This is a welcome addition to the paper. However, I would have liked to see a stronger mechanistic link. First of all, the decision to investigate y321 neurons is surprising, as the authors cite previous work that shows these neurons are ‘implicated in visually-mediated social behavior’ (line 340). In their first figure, they show that oxytocin neurons do not respond to visual social cues. Therefore, the choice to focus on this population seems surprising.

We thank the reviewer for their feedback. Our focus was drawn to the telencephalon because it is known to house GABAergic populations downstream OB input. This was suggestive, as excitatory OB odor responses might inhibit OXT neurons via GABA intermediaries. After imaging GABAergic neurons in the telencephalon (Figure S4-5), we identified that kin water induced activation of subpallial GABA neurons, and hence sought out Gal4 lines that labeled that region, y321 being the most promising option in the existing literature given previous reports of its role in social behavior.

We also believe our choice of y321 is not particularly surprising given that visual input does produce a modest but significant effect on OXT neuron activity (Figure 1c, Figure S1a). Visual social cues slightly reduced OXT activity in the OXT_{PO} subpopulation and significantly reduced activity in the OXT_{PT} subpopulation. While we found no effect of

visual cues on the feeding behavior of isolated fish (Figure 6c), this does not rule out the possibility that visual cue-mediated modulation of y321 neurons can modulate other behaviors, whether it is defensive behavior or social orienting as described by Stednitz et al (2018) (Stednitz et al., 2018).

Second, I would have liked to see an experiment like the one in Figure 2a,b but with suppression of y321 neurons – that would indicate if the suppression of OT cells by kin cue requires y321 activity.

This experiment would indeed be informative if possible, but is technically challenging for various reasons, hence we have not yet attempted it. For example, in our optogenetic experiments, laser illumination was targeted to the subpallium to minimize direct ChR2 activation of the OXT population. In this case, only a subset of subpallial y321 neurons was activated by the laser (Figure S6), but were sufficient to modulate OXT activity. We anticipate a similar issue in optogenetic suppression experiments, with incomplete suppression potentially confounding the conclusions of such an experiment. Further, existing halorhodopsins / light-gated anion channels require yellow-green light activation, which would interfere even more with calcium imaging in our current setup. Overall, we believe the activation experiment that was presented is a sufficient demonstration of circuit connectivity.

Nevertheless, the current version does give a more mechanistic understanding of social buffering in fish. Given this, I really think the feeding data in Fig. 6 belong to a separate manuscript.

We appreciate the reviewer's feedback and understanding. As discussed also in our response to Reviewer 1, we agree with the reviewer that the feeding experiments can be further expanded upon, but they also highlight an important concept that OXT does not simply mediate a single behavior but has multiple outputs, and helps tie our understanding of zebrafish OXT with those of mammals including humans.

Moreover, I have some additional comments related to figures:

Analysis in Fig. 1c and d :

Why isn't the pERK/tERK used here? It would make more sense I think to use that ratio for normalizing to the control group, given how variable staining can be from specimen to specimen.

We thank the reviewer for pointing this out. Indeed, the pERK/tERK ratio is generally utilized in MAP-mapping experiments so that whole-brain maps can be generated without biasing towards brain regions of higher general ERK expression and to account for differences in staining intensity between animals, groups or datasets. In such experiments, images are heavily downsampled with each voxel corresponding to multiple cells. However, tERK is membrane-localized (unlike pERK which is intracellular), hence

in our high-resolution imaging experiments the intracellular tERK signal is absent and gives “division by zero / noise” errors, and would not be useful as a staining control.

An example of high-resolution tERK staining is shown below - note the lack of staining in the intracellular space:

Also, the statistical analysis does not seem to be corrected – please correct for multiple comparison, for example FDR correction.

We originally did not perform multiple comparisons corrections as these were planned comparisons rather than a hypothesis-free analysis. To satisfy the reviewer we have re-done our statistical analysis using a Kruskal Wallis Test with Tukey-Kramer Correction for multiple comparisons.

Fig. 2b,c,d – I only see descriptive stats for these plots, there is no significance analysis as far as I can tell. I believe at least one of the reviewers required proper stats for this figure

We have performed statistical analysis for Figures 2b and d. As 2c is a visualization of spatial distribution, we did not perform statistics for this figure, but the results are quantified in 2d. Multiple comparisons correction was also performed using Kruskal Wallis Test with Tukey-Kramer Correction.

Fig. 2e is not a clear representation of data. If anything, it's confusing. Why not use pie charts or Venn diagrams to more clearly show the overlap?

We agree with the reviewer that the data represented in Figure 2e are not easy to represent in an easily digestible manner. We tried to depict the same data as pie charts (see below) but we felt that this does not really constitute an improvement. The advantage of the current representation is that the response of each neuron to each stimulus can be

compared side-by-side. To make things easier to grasp we have removed the last column in each section. We hope that this resolves the issue.

Fig. 4c: how do we know the oxytocin fibers are dendrites and not axons?

The morphology and projections of individual OXT neurons were previously described by Herget et al (2017) (Herget et al., 2017). A subset of the OXT population sends processes into the telencephalon - these neurons have a pseudo-unipolar morphology where the short processes project anteriorly to the telencephalon and the longer, thinner processes project posteriorly. The diameters, branching patterns and relative lengths of these processes indicate that the dendritic compartment projects to the telencephalon and axonal compartment projects to the hindbrain or posterior areas of hypothalamus. Moreover, that laser illumination targeted to the telencephalon suppresses OXT activity is not consistent with the notion that the anterior processes are axons. In a previous publication, optogenetic activation of OXT neurons leads to calcium activity in hindbrain premotor neurons, further suggesting that the descending projections are axons rather than dendrites (Wee et al, 2019).

Fig. 3e: it seems that clusters 3,4,7, have increased responses to both 'kin' and 'adult'. However, the oxytocin cells have distinct responses to these cues. How would the circuit proposed work to produce these distinct responses?

Figure 4e: The underlying circuit mechanism is indeed an interesting question. The significant fraction of neurons in the subpallium that show correlated responses to kin and adult water (clusters 3, 4 and 7) suggests that this higher order brain region likely serves as a hub for social processing, but not solely via OXT neurons. According to our model, neurons that show related activity patterns between kin and adult odors would not preferential project to the OXT population, but would send their projections elsewhere. There may also be parallel excitatory pathways from within the y321 population by which adult odors may converge onto OXT neurons (e.g. in Figures S4a-b, non-GABAergic neurons were also activated by adult odors).

Fig. 3h: please specify the statistical test used.

Figure 4h: We have indicated the statistical test (two-sided Wilcoxon Rank Sum Test) more clearly in the legends.

What do the authors mean by this phrase: ‘Across a range of thresholds, the fraction of OXT-positive neurons whose activities were suppressed by larval kin- conditioned water was consistently higher than the fraction of neurons that were activated’? what is that range of thresholds exactly?

We apologize for the lack of clarity. The figure we were referring to is Supplementary Figure 2c (pasted below). We plotted the ratios of neurons classified as showing “suppressed” versus “enhanced” activity, as a function of the difference between the mean response of the neuron to a social cue from its response to water flow. The exact range of thresholds evaluated were: 0, 0.05, 0.1, 0.15, 0.2, 0.25, 0.3, 0.35, 0.4, 0.45, 0.5. The consistency of our results across thresholds, namely that kin water induced a higher percentage of suppressed neurons, and adult water a higher percentage of enhanced neurons, indicates that our choice of the threshold 0.05, while arbitrary, is reflecting the general pattern of OXT neuron responses to these different cues (i.e. changing the threshold to 0.1 or 0.2 would not affect the conclusion of our results). We have elaborated on this analysis in the corresponding figure legend.

What other epochs do the authors refer to here: ‘Similarly, in 8-10 dpf tethered fish, kin water significantly reduced the frequency of large-angle swims in tethered fish after kin cue delivery and TRPA1 stimulation but not during other epochs (Figure 5a, Supplementary Figure 8a).’? I hope they don’t refer to ‘pre-cue’ as that is not a logical

statement, right? My understanding is that the cue represents the kin water, so I would not expect to see an effect of kin water before this is given.... Am I missing something here?

The reviewer is indeed right that we do not expect to see an effect of kin water as this is before the kin cue was given, hence it was meant to be used as a negative control where one should not expect to see any differences between kin water and water cues. Further, since the alternating cues are delivered multiple times to the same fish, a pre-cue condition is in fact a post-cue condition at the end of the 5 minute inter-stimulus interval, hence after the first or second stimulus the fish would have in fact had a prior exposure to kin cue. Our analysis has thus demonstrated that this prior exposure does not obviously affect behavioral responses after a long washout period.

Fig. 5f: why was a one-sided test used here? Shouldn't this be two-sided?

The original rationale for a one-sided test was that we had a specific hypothesis for the direction of action. However, we have now used a two-sided test as we agree with the reviewer that it is more appropriate.

For all figures, please make the font size within panels bigger.

We have increased the font size to the maximum specified by journal requirements.

REFERENCES:

Herget, U., Gutierrez-Triana, J.A., Salazar Thula, O., Knerr, B., and Ryu, S. (2017). Single-Cell Reconstruction of Oxytocinergic Neurons Reveals Separate Hypophysiotropic and Enkephalotrophic Subtypes in Larval Zebrafish. *Eneuro* 4, ENEURO.0278–16.2016.

Macosko, E.Z., Pokala, N., Feinberg, E.H., Chalasani, S.H., Butcher, R.A., Clardy, J., and Bargmann, C.I. (2009). A hub-and-spoke circuit drives pheromone attraction and social behaviour in *C. elegans*. *Nature* 458, 1171–1175.

Stednitz, S.J., McDermott, E.M., Ncube, D., Tallafuss, A., Eisen, J.S., and Washbourne, P. (2018). Forebrain Control of Behaviorally Driven Social Orienting in Zebrafish. *Curr. Biol.* 28, 2445–2451.e3.

REVIEWERS' COMMENTS

Reviewer #3 (Remarks to the Author):

The authors have addressed all of my concerns. I would recommend publication of this manuscript in Nature Communications.

I only have two additional comments on some minor errors:

Supplementary Figure 2d & e, df/f scale. The df/f scale should range from -0.3 to 0.3. Please fix.

Supplementary Figure 6b, bottom. The way anatomical axes are annotated here is confusing because "L" in the dorsal view can be interpreted as either "left" or "lateral". I'd suggest the authors either use L (lateral) for both views or clearly explain these abbreviations in figure legends.

Yunlu Zhu, Ph.D.

Reviewer #4 (Remarks to the Author):

The authors addressed my comments

Response to Reviewer #3:

I only have two additional comments on some minor errors:

Supplementary Figure 2d & e, df/f scale. The df/f scale should range from -0.3 to 0.3. Please fix.

Thank you for spotting this error, we have fixed it accordingly

Supplementary Figure 6b, bottom. The way anatomical axes are annotated here is confusing because "L" in the dorsal view can be interpreted as either "left" or "lateral". I'd suggest the authors either use L (lateral) for both views or clearly explain these abbreviations in figure legends.

Thank you for your suggestion. We have decided to stick to L = left and R = right, and have explained the abbreviations in the legends accordingly.